# Development of KOSEN Weather Station and Provision of Weather Information to Farmers

**DOI:** 10.3390/s22062108

**Published:** 2022-03-09

**Authors:** Jeyeon Kim, Daichi Minagawa, Daiki Saito, Shinichiro Hoshina, Kazuya Kanda

**Affiliations:** 1Department of Creative Engineering, National Institute of Technology, Tsuruoka College, Tsuruoka 997-8511, Yamagata, Japan; hoshina@tsuruoka-nct.ac.jp (S.H.); kanda@tsuruoka-nct.ac.jp (K.K.); 2Advanced Engineering Course at National Institute of Technology, Tsuruoka College, Tsuruoka 997-8511, Yamagata, Japan; s200010@edu.tsuruoka-nct.ac.jp (D.M.); s200006@edu.tsuruoka-nct.ac.jp (D.S.)

**Keywords:** agriculture ICT, weather station, could server, Aßmann’s aspiration psychrometer, WioLTE, alert notification

## Abstract

In recent years, environmental information monitoring in the agricultural field has become an important issue. There is an increasing demand for meteorological information in local areas such as a rice field, a greenhouse, etc., owned by an agricultural worker. Conventional research has been actively conducted on weather stations in local areas. However, weather stations that are inexpensive, highly accurate, and have achieved stable measurements indoors and outdoors for long periods of time (over a year) are not reported. In addition, there is a lack of research that simultaneously acquires weather information, stores weather information, and provides weather information to farmers. These three functions are important in the agricultural field. In this paper, we discuss the development of a meteorological observation device, the construction of a cloud server for storing meteorological information, and the provision of information to users. First, we develop the novel meteorological observation device (KOSEN-Weather Station), which applies a simple Aßmann’s aspiration psychrometer for highly accurate temperature and humidity measurements. To evaluate the reliability of KOSEN-WS, we compare the weather information measured by KOSEN-WS with that of WXT520. As a result, it is shown that KOSEN-WS is viable. Then, KOSEN-WS is installed in the field, and the stability and durability of KOSEN-WS are examined. As a result, the KOSEN-WS has been operating stably over 19 months and provides weather information to users. Then, it is shown that the KOSEN-WS is able to operate continuously under the environment of −16.5 °C to 44.9 °C. Next, for the storage of meteorological information, we construct the cloud server. Then, a webpage is created to provide easy-to-understand weather information to farmers. Furthermore, to prevent damage to crops, if the current temperature is lower than the set temperature, or if the current temperature is higher than the set temperature, an alert is sent to the farmers. As a result, the system is highly evaluated by agricultural workers and JA staff. From the above results, the effectiveness of this system is shown.

## 1. Introduction

The situation of agriculture in Japan has many problems, such as the aging of farmers, the shortage of labor due to the decrease in the working population, and the difficulty of obtaining new farmers [1,2]. In Japan, Information and Communication Technology (ICT) in agriculture has been promoted since the 1990s, and many research institutes and companies have entered the ICT field in agriculture. In order to solve these problems, “smart agriculture” has been attracting attention [3,4]. Smart agriculture refers to the use of technologies such as robotics and ICT on farms to achieve labor savings, high-quality production, and so on. The introduction of smart agriculture can solve the problems of labor savings, labor shortages, and difficulty in finding new farmers.

To build a decision support system for new farmers and young farmers, it is necessary to have information on the growing environment, crop growth, agricultural work, and so on. These data will need to be visualized and analyzed using Artificial Intelligence (AI). Nonetheless, due to the wide variety of agricultural fields, we are focusing on meteorological information, which is one type of growth environment information. Crop growth information and a work diary are collected along with various environmental information (temperature, humidity, water level, soil moisture, water temperature, etc.) in the field. Then, by combining the information with the know-how of experts, it is possible to support decision-making such as pesticide spraying and harvest times. Decision-making support for agricultural followers can improve productivity, cope with climate variation, and support new entrants. In order to achieve these goals, weather information acquisition, weather information storage, and weather information provision to users are important elemental technologies. Moreover, environmental information monitoring according to the purpose is an important issue in the agricultural field. In particular, the stable acquisition of weather information is an important issue for productivity improvement, countermeasures against damage caused by the weather, and other purposes. Moreover, requirements are increasing for weather information in local areas (e.g., each rice field and each greenhouse) to improve productivity.

A conventional meteorological observation system is described. Japan has a regional meteorological observation system, the Automated Meteorological Data Acquisition System (AMeDAS) [5], that observes meteorological data at regional representative points. At the 840 AMeDAS locations throughout Japan, the system automatically observes meteorological data such as temperature, wind direction, wind speed, and sunshine duration. Because the meteorological data observed by AMeDAS are data of representative points of a wide range, grasping the meteorological conditions of each field is difficult. The Agro-Meteorological Grid Square Data [6] developed by the National Agriculture and Food Research Organization (NARO) is a meteorological data service system aimed at making effective use of meteorological information at agricultural fields. Daily nationwide meteorological data can be provided on demand in units of approximately 1 km^2^ (reference area mesh). However, the meteorological data are based on simulation. They are not actually measured data. It is difficult for these meteorological observation systems to acquire meteorological data for each area, although such data have been increasingly required in recent years. These systems are unable to precisely predict gusty winds, tornadoes, guerrilla rains, unexpectedly strong rains, sudden rains, etc., in the local area at present.

Various studies on the acquisition of meteorological information in the local area have been actively conducted to solve these problems [7,8,9,10,11,12,13,14,15,16,17,18,19,20,21,22,23,24,25,26,27,28,29,30,31,32,33,34,35,36,37,38,39,40]. Nevertheless, there has been insufficient research on WSs that are inexpensive and can acquire weather information stably over a long period of time. The details are described in Section 2.

To overcome the difficulties described above, this paper describes the development of a novel weather station that is inexpensive and durable, and the construction of a cloud server to provide weather information and alert notifications to farmers. The contributions of the study are as follows:Development of weather station

The WS is inexpensive and easy to install. In addition, a simplified Aßmann’s aspiration psychrometer for highly accurate temperature and humidity measurements is developed. A WS that can perform measurements stably over a long period of time is necessary.

Cloud server construction.The server stores weather information.Provision of weather information.

We create a webpage where users can check weather information at all times and all places and provide weather information for farmers that is easy for users to understand. Then, alert notifications provided to users at high or low temperatures to prevent damage to crops are examined.

Section 2 describes related work on collecting meteorological information in local areas, and Section 3 presents the KOSEN-WS development and cloud server construction. Section 4 and Section 5 describe the basic experiments and field experiments for proving the reliability, stability, and durability of the KOSEN-WS. Section 6 presents the provision of weather information to the farmers. Finally, the conclusions are presented in Section 7.

## 2. Related Work and Methodology

### 2.1. Related Work Methodology

In recent years, requirements for weather information in local areas have increased to improve productivity and to carry out labor saving. Furthermore, various meteorological observation devices capable of acquiring weather information in local areas have been developed. These devices are described herein.

The WXT520 Series Weather Transmitter [7] is an integrated meteorological instrument that can observe meteorological data of six kinds. It is possible to connect an external sensor to configure a meteorological observation hub. The WXT520 can acquire weather information stably. However, it is necessary to use an AC power supply because of its power consumption and the price is high. The field server developed by Fukatsu et al. [8] is equipped with various environment sensors and a camera. It can access a field server from a remote location via the Internet using a 3G line or wireless LAN. Furthermore, it is possible to check and collect environmental information. In addition, because the acquired weather information can be transferred to a web server, it is accessible from anywhere using a web browser. Nakayama et al. [9] conducted a survey of the required specifications of a server-installed type field server installed in the field and determined appropriate specifications of a field server available in the field. However, because the server-installed type field server consumes large amounts of power, using the field server is difficult in places where securing a power source is difficult. To observe the paddy rice growth process and the increase in vegetation coverage, Sekiguchi et al. [10] used a web camera. Nevertheless, the system is useful in places where power can be secured. e-kakashi [11], a cultivation navigation tool produced by SoftBank Corp., provides a service to acquire field environmental data such as temperature, humidity, geothermal heat, and amount of solar radiation. It also analyzes the growth status. Nevertheless, the equipment is expensive. The e-kakashi gateway requires an AC power supply (100V). In the literature [12], an affordable and accurate weather monitoring system has been developed. A comparison was made between the developed ATMOS41 and Vaisala’s weather station to show the effectiveness of the developed system. Nonetheless, the authors have not evaluated the performance of the proposed system in terms of its ability to acquire weather data stably over a long period of time. In addition, the price of the system is not mentioned, although it is said to be affordable.

Aiming to solve the above problems, various studies have been actively conducted on power-saving and inexpensive WSs. Table 1 summarizes conventional research on power-saving and inexpensive WSs. Here, “x” in the measurement period column means that the measurement period is not mentioned.

In the WSs [13,14,15,16,17,18,19] using the Raspberry Pi, WSs [20,21,22,23,24,25,26,27,28,29,30] using the Arduino, and WSs [35,36,37,38,39] using other microcontrollers, each microcontroller plays the roles of both a sensor for measurement and a gateway. In the WSs [31,32,33,34] using Arduino and Raspberry Pi, the Arduino performs the sensor measurements, and the Raspberry Pi acts as the gateway. The measured data are transmitted to a server or PC using communication methods such as 3G/4G, WiFi, Zigbee, and LoRa. The sensors include sensors for temperature, humidity, soil moisture, wind speed, wind direction, and rainfall.

The power consumption and communication ranges are also discussed. WSs using 3G/4G [13,14,20,21,31,35,36] or WiFi [15,16,22,23,24,32,37] may be difficult to use in places where a power supply cannot be secured, owing to their high power consumption. WSs using Zigbee [18,25,26,27,33], LoRa [19,28,29,30], SigFox [38], and Bluetooth [39] are advantageous in places where power supplies cannot be secured, owing to their power-saving features.

The 3G/4G network has a large communication range, so sensor data can be sent to a server over a wide range. However, WiFi, Zigbee, LoRa, SigFox, and Bluetooth have a smaller communication range than 3G/4G networks, and require a nearby router or repeater to connect to the Internet. Nevertheless, in the field, radio waves may be blocked by crop growth, and the communication range can become narrower [40]. In many cases, it is not possible to connect to the nearby Internet in the field. These communication methods are often shielded or USB types that connect to a microcontroller. Furthermore, shield-type (Zigbee, LoRa, SigFox, etc.) and USB-type (3G/4G) communication devices for connecting to microcomputers can easily be corroded by moisture.

The measurement period and durability of a WS are important, as it must be operated continuously for a long period of time. In particular, it is necessary to study the durability in mid-summer and mid-winter. References [24,30,31,32,34,37,39] describe the measurement period of the WS as within one day, and references [21,26,27] describe the measurement period of the WS as within one week. These studies only show the operation checks of the developed WSs. Reference [22] studied the reliability and durability of a system by continuously measuring for more than three months. However, as the growth time of crops is much longer, a study over a longer period of time is necessary. In particular, the durability of these systems during mid-summer and mid-winter has not been studied. The provision of weather information has also been explained, as described in the literature [18,21,22,24,26,27,28,30,31,34,37,39]. By contrast, in these references, the weather information is presented to confirm the operation of the WS; there is insufficient consideration of the information presentation, i.e., making the information easy to understand for farmers.

Based on the explanation presented above, this paper presents a simultaneous discussion of the WS development, server construction, and weather information provision. To develop a WS that is useful by small-scale and medium-scale farmers, we consider the sensor, microcontroller for sensor measurement, communication function, circuit board design, power supply, power consumption, WS size, and price. To store the weather information uploaded from the WS and to provide it to farmers thereafter, we must also build a server. This system has been developed according to an engineering design for which the details are described in the next section.

### 2.2. Methodology

We describe the methodology of engineering design to develop the system. This methodology is based on the literature [41,42,43].

Problem definition: Meteorological information is applicable to the monitoring of growth, prediction of harvest time, prevention of pests and diseases, dealing with climate change and abnormal weather, and prediction of frost damage. The weather information acquisition, server construction, and provision of weather information to farmers are important elements of work in agricultural fields. Nevertheless, studies of WSs that are inexpensive, precise, and capable of obtaining stable weather information over a long period of time in local areas are lacking. Moreover, studies of the provision of reliable and easy-to-understand weather information needed by farmers are insufficient, as are studies examining alert notifications at high or low temperatures. In particular, no reported study has considered the three elements above simultaneously.Opinions that include the approach: To grasp the current situation in agricultural fields, we conducted literature searches and field visits. We also administered a questionnaire survey to farmers and JA staff members about the types of weather information necessary for agricultural work and the weather information that should be provided to users. Questionnaire items for WS include the crop type, the type of weather information required (sensor type), the period of use, the price, the size, and the installation location. Based on questionnaire survey results and the knowledge of the authors, the specifications used to implement the WS were determined. The questionnaire items related to information provision include the crop types, pull-type weather information, push-type weather information, frequently used places, and devices for providing information. After summarizing the WS specifications, server construction method, and information provision method based on the results of the questionnaire survey and the authors’ knowledge, methods and steps to resolve difficulties in the next “instrument” are discussed.Instrument: The WS, server, and information provision are described hereinafter in that order. The questionnaire survey results and the authors’ knowledge were summarized to support the development of a WS that can acquire weather information stably over a long period of time. We describe the various components, including sensors, microcomputers, communication devices, circuit boards, and power supplies, in addition to power consumption, WS size, and server construction. Weather information necessary for agricultural work was investigated. Subsequently, we considered the measurement precision, price, and measurement range of sensors for weather information of each type. We examined whether the microcomputer can measure various sensors easily, whether the microcomputer can operate stably for a long time, and so on. To ensure stable transmission of meteorological information to the server at any location, one must consider the communication device and the connection method between the microcomputer and the communication device. We discuss the circuit board design. A durable circuit board that is useful for a long time in indoor and outdoor environments must be designed. We discussed the WS power consumption. One must consider where power is secured and where it is not. In places where power is secured, we can use a USB adapter. However, in places where power is not secured, solar panels and batteries must be considered for use to mitigate some of the power consumption of a WS. Because Japan has a rainy season, securing a power source that is useful during this period is necessary. The WS size must be sufficiently small to be installed easily by farmers. In addition, after WS is installed, the WS should not interfere with farm work after installation. We discuss items to be considered when building a server. Building a server to accumulate weather information uploaded from the WS and to provide weather information to users is necessary. There are servers of two types: on-premises and cloud servers. Moreover, the initial cost and running cost must be considered. Information provision to farmers is described. Results of the questionnaire survey, field visits, and the authors’ knowledge based on literature research are summarized. In particular, devices for providing information, the kinds of pull-type information, and the kinds of push-type information in weather information provision are summarized. After summarization of the points above, we built a prototype of the WS, constructed a server, created a webpage for providing information, and provided alerts.Validation: For WS validation, we conducted basic and field experiments. By basic experimentation, we validated whether this prototype is useful in a real environment. The WS and the reference WS were installed at the same place. The precision of the sensors was compared. In case of WS failure, the causes were listed. Countermeasures were taken. We resolved the difficulties that occurred in the basic experiment and conducted the field experiment, which is a hot and humid environment. Using field experiments, we investigated whether the WS can operate stably and continuously for a long time in the real environment. We conducted field experiments in four places (fields and greenhouses with power supply, and fields and greenhouses without power supply) to verify whether the WS can operate for a long time stably and continuously. Then, in cases of WS failure, the causes were listed. Countermeasures were taken. For validation of the information provision, we set up WSs in the fields and validated whether the pull-type and push-type weather information can be provided reliably to farmers and JA staff. The validation method involved a questionnaire survey among agricultural workers and JA staff. For the pull-type information provision, we summarized the readability of the webpage, types of weather information, text size, and graph size. In the case of push-type information provision, we summarized whether alert notifications were sent reliably, whether alert notifications continued to be sent until users confirmed them, how to stop alert notifications, and how to reset alert notifications. We made a list of improvements from the questionnaire survey and solved them by repeating the same process as in “Instruments” again until all the improvements were clarified.Synthesis/analysis process: In cases of WS failure, we retrieved them, analyzed the causes of failure from both hardware and software aspects, and resolved the related difficulties. During the basic experiment, we conducted the operation check of the prototype and analyzed hardware and software difficulties. For hardware issues, we analyzed the sensor precision, the soldering of the board, and the solar panel and battery capacity in places where power cannot be secured. Then, we discussed solutions. For software issues, we analyzed the microcontroller program, sensor measurement by the microcontroller, and whether or not the measurement data can be uploaded to a server, and discussed solutions. In the field experiment, we analyzed hardware and software problems to verify that the WS can operate stably and continuously for a long time. Hardware difficulties included failures of each sensor, short-circuit generation because of overcurrent, and corrosion and peeling off of soldered joints in a hot and humid environment. Software difficulties included the occurrence of overflow and microcontroller shutdown because of sensor failure. The solutions to these difficulties were discussed. Considering the questionnaire survey results and the authors’ knowledge, the weather information provision and alert notifications were improved. The problems described above were solved by repeating the same process as in “Instrument” and “Validation”.

## 3. KOSEN Weather Station and Provision of Weather Information to Farmers

### 3.1. System Overview

The acquisition, storage, and provision of weather information are an important elemental technology that form the platform in the agricultural field. The system in this study is an example of an embodiment of such a platform in the agriculture field. Figure 1 presents an overview of this system, which portrays the entire process from meteorological measurements to weather information provision to farmers. Specifically, the KOSEN Weather Station (KOSEN-WS) periodically acquires weather information in local areas such as rice fields and greenhouses and transfers the weather data to a cloud server. Then, the cloud server accumulates meteorological data and provides users (agricultural workers) with the necessary information.

The KOSEN-WS includes a sensor unit, a power supply, and a gateway. Details of KOSEN-WS are presented in Section 3.2. The cloud server comprises a web server (Nginx), an API server (tomcat), and a SQL server to provide information to users. Details of the cloud server are described in Section 3.3.

The reason that we named the weather station KOSEN-WS will be expressed. The English full name of our college is “National Institute of Technology (NIT), Tsuruoka College”. In Japanese, the NIT is called KOSEN, and our college is called Tsuruoka KOSEN. Thus, we named the weather station as KOSEN-WS.

The merits of this system are that it is inexpensive, easy to switch between power sources (commercial power and solar panels), can be used for long periods of time, and can provide farmers with easy-to-understand weather information.

### 3.2. KOSEN Weather Station

Figure 2 presents the composition of KOSEN-WS. KOSEN-WS consists of a sensor unit, a power supply, and a gateway. The sensor unit periodically acquires weather information. The power supply provides power to the sensor unit and the gateway. Then, the gateway transfers the weather information obtained from the sensor unit to the cloud server.

First, the sensor unit is described. The sensor unit consists of a microcontroller (Arduino Pro mini; SparkFun Electronics^®^, Niwot, CO, USA) and various sensors. Six types of sensors, for temperature, humidity, solar radiation, wind direction, wind speed, and rainfall, can be connected to the sensor unit. In addition, only the necessary sensors can be connected to the sensor unit as needed. The sensor unit periodically acquires weather data from the connected sensors.

As described in this paper, the types of weather information to be provided to farmers were ascertained through a questionnaire survey of farmers and JA staff; six types of weather information were determined. The sensors used for this study are described below. The temperature/humidity sensor (SHT-21; Sensirion Corp., Tokyo, Japan) is used. This temperature/humidity sensor is commonly used. We have developed a simplified version of Aßmann’s aspiration psychrometer for highly accurate temperature and humidity measurements. Aßmann’s aspiration psychrometer for atmospheric humidity measurements was developed between 1886 and 1889 and is still recommended as a working-standard instrument today [44]. Aßmann’s aspiration psychrometer can measure temperature and humidity with high accuracy, but it is expensive. In this paper, we have developed a simplified version of Aßmann’s aspiration psychrometer. As shown in Figure 3, this device consists of a ventilation pipe, a small fan, a fan attachment, and a temperature/humidity sensor. A temperature/humidity sensor is placed in the ventilation pipe. Then, highly accurate temperature/humidity measurement was achieved by blowing air through a small fan for 30 s.

Wind/Rain Sensor Assembly [45] is used for wind speed, wind direction, and rainfall. This Wind/Rain Sensor Assembly is inexpensive and easy to install. The amount of solar radiation is measured by using a solar panel.

The circuit board design is described. The circuit diagram and circuit board for the sensor unit are portrayed in Figure 4. A durable circuit board that is useful for long periods of time in hot and humid indoor and outdoor environments must be designed. Printed circuit boards (PCBs) and printed circuit board assemblies (PCBAs) are recommended for stable measurement over a long period of time. Actually, PCBs are inexpensive, but they require soldering, and require countermeasures against corrosion, etc. Although PCBAs are more expensive, they require no soldering and are more durable. Eventually, PCBs were chosen because of their price, particularly considering that farmers operating on a small scale or medium scale use them. Furthermore, the wiring was designed to be sufficiently thick to withstand overcurrent, etc.

Next, the gateway periodically transfers the weather information measured from the sensor unit to the cloud server. The microcontroller used for the gateway is WioLTE [46], which is installed with an LTE communication module as standard equipment. Consideration of connection methods between the microcomputer and the communication device is necessary. In general, the most common method is connection to the microcomputer using a USB connection or a shield. Many microcomputers also have a built-in communication function. It is necessary to consider compatibility between communication devices and the microcomputer, and corrosion in high-temperature and humid environments. Microcomputers with built-in communication functions are more expensive, but they can operate stably and continuously. They are useful for a long time, even in hot and humid environments. We discussed communication systems. The most used communication systems are cellular networks and Low-Power Wide-Area (LPWA) networks. Cellular networks (4G and 3G) have a wide communication range and a low initial cost. However, communication devices using cellular networks consume much power and have running costs. LPWAs include ZigBee, LoRa, Sigfox, and so on. Communication devices using LPWA have low power consumption and low running costs. However, LPWA might cause communication instability because of crop growth. In addition, the initial cost is highly attributable to the need for a repeater nearby to transmit weather information to the server.

The WS size should be less than 200 mm wide, less than 300 mm high, and less than 150 mm deep because the WS size should be sufficiently small for farmers to install easily.

Finally, the power supply is described. To use the WS for a long time, even outdoors, low power consumption is preferred. In general, the microcomputer power consumption is highest during communication. Supplying sufficient power is necessary, even when using cellular networks (4G and 3G). One must also consider where power can be secured and where it is unavailable. In places where power is available, we can use a USB adapter. By contrast, in places where power is not available, solar panels and batteries must be used considering the power consumption of a WS. Because Japan has a rainy season, it is necessary to secure a power source that is useful during this period. The amount of power generated by the solar panel and the battery capacity must also be considered. Furthermore, this WS can switch easily between the two power sources. In places where the power supply cannot be secured, the WS uses a power controller [47] for solar panels and automobile batteries. 

A questionnaire survey was administered among workers on small-scale and medium-scale farms and JA staff on WS prices. It elicited various opinions from agricultural workers. However, after discussing the commercialization of this system, we came to the conclusion that the initial cost of this WS is expected to be less than USD 1000 and that the running cost is expected to be less than USD 10. 

Figure 5 presents an example of a KOSEN-WS installed in a place where no power supply can be secured. It comprises six sensors, a solar panel, and a battery. Figure 6 shows an example of a weather station set up in a house where power can be secured. If a commercial power supply is used, a battery controller is required instead of a power supply controller to use LiPo batteries [48]. Moreover, when installing the KOSEN-WS in a greenhouse, sensors for rainfall, wind speed, and wind direction are not required. KOSEN-WS is inexpensive and can acquire meteorological information stably. The box for KOSEN-WS is waterproof, so there is no water penetration by rain. In addition, by using a waterproof spray, it can be used both outdoors and in a humid greenhouse. The battery for the solar panel was placed on the ground in this paper, but it is better to place the battery in the soil for longer use.

### 3.3. Cloud Server

The cloud server stores weather data transferred from KOSEN-WS and provides information to farmers. Cloud servers such as Azure [49] and AWS [50] are available, but Azure is used for this study. Figure 7 shows an overview of the cloud server. KOSEN-WS forwards the measured weather data to the webserver. The transmitted weather data are then stored in the database server via the application (AP) server. The user can check the weather information by accessing the webpage when necessary. Moreover, alert notifications are provided by the AP server through LINE. Users can then set the temperature setting for alert notification through LINE.

Nginx [51] is used as the webserver to process HTTP. Since Nginx is designed to handle a large volume of communication, the processing speed is difficult to slow down, even if many clients exist. For the AP server that processes HTTP requests, Tomcat [52] and Java are used. The latter can use JAX-RX, a library for implementing RESTful API [53]. MySQL [54], which is available as an open source, is used as the database management system (DBMS) for the DB server that stores data. In addition, the Messaging API [55] provided by LINE is used to implement the alert function.

The device for providing weather information to farmers is not a dedicated device, but is rather a smartphone that is always carried by farmers. Dedicated devices can provide good information, but they are expensive and take time to develop. The application for providing weather information was then chosen to be LINE, an application that is already installed on the smartphones of agricultural workers and is often used by them. Pull-type and push-type modes are available for information provision. The pull-type information provision is a method that allows farmers to access a webpage and browse weather information when users want to check it. The pull-type information items are temperature, humidity, solar radiation, wind speed, wind direction, and rainfall. However, for indoors (greenhouses), only temperature, humidity, and solar radiation are provided because wind speed, wind direction, and rainfall are not necessary. Weather information for the past (one week) is also provided as needed. Push-type information includes alert notifications. The push-type information provision is a method that presents information from a server, irrespective of the agricultural worker’s intention. For example, an alert notification is sent irrespective of the user’s intention when the weather conditions can cause damage to the crop. Using the cloud server entails a running cost. It costs approximately USD 10 or less on average per WS.

The user can check the weather data by accessing the webpage. Figure 8 is the home page for users. The webpage displays the temperature and humidity that farmers frequently check by default. Details are presented in Section 6.

This paper simultaneously discusses the WS development, server construction, and weather information provision. In order to develop a WS that can be used by small- and medium-scale farmers, we consider the sensor, microcontroller for sensor measurement, communication function, circuit board design, power supply, power consumption, size of the WS, and price. We also need to build a server to store the weather information uploaded from the WS and provide it to farmers. This system has been developed according to an engineering design. The details are described in the next section.

## 4. Basic Experiments

### 4.1. Experiment Method

To evaluate the KOSEN-WS reliability, the precision of the meteorological data measured by KOSEN-WS is evaluated. Specifically, a comparative experiment was conducted to evaluate KOSEN-WS and WXT520 [6]. The experimental site was the roof of the college building from 19 to 26 November 2015. Figure 9 shows the installation location of KOSEN-WS and WXT520. The height of the airflow meters is the same and the distance between the two devices is approximately 4 m. The evaluation items are meteorological data such as temperature, humidity, wind speed, rainfall, and solar radiation.

### 4.2. Experimental Results

The meteorological data measured by KOSEN-WS are compared with the meteorological data measured by WXT520.

Figure 10 presents the results of the comparison of temperature data. It is apparent that the temperature data of KOSEN-WS and temperature data of WXT520 have almost identical results. These results show that the simple Aßmann’s aspiration psychrometer is valid.

Figure 11 shows the comparison results of humidity. The difference between KOSEN-WS and the WXT520 was approximately 17% on average. This difference occurred due to the use of an inexpensive sensor. Subtracting 17% from the humidity data measured by KOSEN-WS shows that the difference is smaller, as shown in Figure 12. In this study, only one week of weather data were used for correction, but it is necessary to collect weather data over a long period of time to clarify the difference between the two sensors. As shown in Figure 10 and Figure 12, these results show that KOSEN-WS for temperature/humidity measurement is effective.

Figure 13 shows the results of the comparison of wind speeds. The overall trend is the same, but there are errors between the two measurement results. The measurement method of this anemometer converts the wind speed using the number of rotations per second. In other words, sampling of less than 1 s is not possible. The KOSEN-WS measures wind speed every second and outputs the average of 5 s, while the WXT520 measures wind speed every ¼ s by default. Because of these factors, an error between the two anemometers occurred.

Figure 14 shows the rainfall results. The difference from WXT520 is large. The rain gauge used in this study is an inexpensive tumbling bucket ammeter, which does not provide good sensor precision. Nevertheless, it is possible to determine whether it is raining or not. The rain gauge must be considered in future work.

Figure 15 presents the results of the comparison of the amount of solar radiation and shows that both sensors have almost identical results.

In this section, to investigate the reliability of KOSEN-WS, a comparative experiment was conducted to evaluate KOSEN-WS and WXT520 [6]. The results showed that KOSEN-WS is fully usable, with good measurement results for temperature, humidity, wind speed, and solar radiation.

## 5. Field Experiments

### 5.1. Experiment Method

To examine the possibility of stable data collection over a long period of time, we evaluate the performance of the KOSEN-WS in terms of stability and durability in the field. In this experiment, the KOSEN-WS uses a commercial power supply in the place where a power supply is available and uses a solar panel and battery in the place where a power supply is not available. Then, we study the operating time of the WS over a long period of time and the available weather conditions.

The experiments were conducted at five locations in Yamagata Prefecture (Tsuruoka, Mikawa, Higashine, Tendo, and Nanyo, Japan). Figure 16 shows the installation of KOSEN-WS in Yamagata Prefecture. Mikawa and Tendo were set up indoors and outdoors in greenhouses where power could be secured. Tsuruoka, Tendo, and Nanyo were set up outdoors where no power source was available. In summer in Yamagata, the temperature inside the greenhouses often reaches 35 °C or higher, and the humidity inside the greenhouses makes it a good place to test their durability. In addition, Yamagata Prefecture has few hours of sunlight in winter. Especially in Tsuruoka, there are only approximately 20 to 50 h of sunlight due to the snow and cloudy weather. If KOSEN-WS, which uses solar panels, can collect stable data during the winter period when sunlight hours are very short, KOSEN-WS can be used anywhere. Furthermore, the installation period was from April 2019 through March 2021.

### 5.2. Experimental Results

To evaluate the stability and durability of the KOSEN-WS, the operating time and the operatable temperature range of the KOSEN-WS in the case of using a commercial power supply and the case of using solar panels and batteries are described. Table 2 presents the operating time and the operatable temperature range of KOSEN-WS when using a commercial power supply.

The KOSEN-WS installed indoors (the tomato growing greenhouse) in Mikawa, where power can be secured, has been in continuous operation for more than 18 months. Meanwhile, the KOSEN-WS installed outdoors, where the power supply can be secured, failed after around nine months of continuous operation. Currently, the KOSEN-WS has been re-installed and is running. The KOSEN-WS installed outdoors in Higashine has also stopped. The reason is that a farmer cut the electric wire during weeding work. Currently, the KOSEN-WS installed outdoors in Higashine has been in continuous operation for more than 10 months since it started working again. The operatable temperature range of KOSEN-WS is described. The maximum temperature of +44.9 °C and the minimum temperature of −10.9 °C were measured. It was confirmed that KOSEN-WS can be used in weather conditions ranging from approximately −11 °C to 45 °C. Figure 17 shows the temperature data measured by the WS installed in Mikawa (indoors). This figure shows the temperature data over a period of 19 months.

Table 3 presents the operating time of KOSEN-WS when using solar panels and a battery. The WS in Tsuruoka operated for six months, the WS in Tendo for three months, and the WS in Nanyo for four months. The WS installed in Tsuruoka is in operation again. Moreover, the WS installed in Tendo has been in operation for three months because it was removed for snow removal. The conditions under which the system can operate at certain temperatures are described. The WS installed in Tsuruoka measured a maximum temperature of +41.3 °C, and the WS installed in Nanyo measured a minimum temperature of −16.5 °C. That is, the KOSEN-WS was able to operate continuously under the environment of −16.5 °C to 44.9 °C. Figure 18 shows the temperature data measured by KOSEN-WS installed in Nanyang. This figure shows the temperature data over a period of 6 months.

The authors conducted a questionnaire survey of farmers and JA staff members regarding sensor precision, WS size, and ease of installation. As a result, they gave good evaluations. We compared the conventional WS with the proposed WS in terms of price, precision, operating time, and ease of installation. In conventional research, inexpensive WS, precise WS, and easy-to-install WS have already been described. However, there is no WS that satisfies all the requirements, such as price, precision, operating time of more than one year, and ease of installation, except for this KOSEN-WS.

### 5.3. Discussion

Here, we will discuss solar panel installation, causes of failure, and power supply. First, the installation method of the solar panels will be discussed. The experimental sites are Tsuruoka, Higashine, Tendo, and Nanyo in Yamagata Prefecture. Yamagata Prefecture is a region with a lot of snow and little sunshine in winter. Especially in Tsuruoka, there is a lot of snow from December to February and only around 26 to 40 h of sunlight. In order to operate the WS even in places where the power supply cannot be secured in winter, it is necessary to save the power of the KOSEN-WS and to have solar panels and batteries that can be sufficiently charged even with little sunlight. The solar panel output used for this study is 100 W. The automobile battery capacity is 100 Ah. Furthermore, in places where the power supply can be secured, the WS uses the power controller [48] for a LiPo battery to supply the power stably. The solar panels need to be installed so that they do not accumulate snow. Figure 19 shows the WS installed in Tsuruoka (outdoors) in winter; it can be seen that there is a lot of snow around the KOSEN-WS, but the solar panel is not covered with snow. The experimental results show that the system is in continuous operation even in winter.

Next, the causes of the WS failure are described. At the beginning of the installation, it sometimes stopped in 2 weeks or 2 months. The cause of the outage was both software and hardware. Causes of program stoppage included counter overflow and reset timing error. There were also cases when the microcontroller stopped because it could not take measurements due to a sensor failure. Counter overflow and reset timing errors were corrected by reviewing the program and correcting the counter. Moreover, as a countermeasure for sensor failures, the microcontroller was able to operate continuously without stopping by returning NaN when the sensor could not be measured. Causes of failure in hardware included component mounting error due to insufficient solder, short-circuit due to overcurrent, and failures due to corrosion. In the case of failure due to short-circuit caused by overcurrent or corrosion, we redesigned the circuit board. Furthermore, in order to use the KOSEN-WS for a long period of time, the circuit board of the KOSEN-WS needs to be converted to a printed circuit board (PCB) assembly. Currently, KOSEN-WS is easily corroded by moisture because we soldered the components. However, it is possible to prevent corrosion due to moisture by using a PCB assembly in the KOSEN-WS.

Finally, one of the difficulties we faced in this experiment was a stable power supply. In many cases, the KOSEN-WS powered by the USB adapter could not operate for a long time. The reason is that microcontrollers may stop working even if the voltage changes for a short time. In particular, when the microcomputer performs LTE communication, the power consumption of the microcontroller suddenly increases and the USB adapter may not be able to support it. In this paper, we used a power controller and a LiPo battery to provide a stable power supply. The power controller can supply power stably from the LiPo battery. All the KOSEN-WS that operated continuously for a long time were those that used the power controller. Even if the external power supply is not available, the LiPo battery used in this paper can be used for around 2.5 days. In the case of using solar panels and batteries, a stable power supply was achieved by using the power controller.

## 6. Provision of Weather Information to Farmers

Here, the provision of weather information and alert notifications to farmers is considered. In particular, “whether it was possible to provide information that is easy to understand for agricultural workers” and “whether it is easy to operate” are considered. Then, the operation of the alert notifications at the time of high temperatures or low temperatures is confirmed. To provide easy-to-understand information through the webpage, we gathered opinions from JA staff and agricultural workers, and produced a webpage. The opinions of agricultural workers and Japan Agricultural Cooperatives Zen-Noh Yamagata (JA) staff were as follows.

The temperature graph should be placed at the top of the screen, as farmers check the temperature most often.Two days of weather information should be presented on a graph.If necessary, data for one week should be presented on a graph.The maximum and minimum temperatures of the previous day, today’s minimum temperature, and current temperature should be presented in numbers.An alert notification interval should be provided.The text should be large enough to be read easily on a smartphone.

These comments were reflected on the website and in the alert notifications. The webpage displays the temperature and humidity that farmers frequently check by default (Figure 20a). For easy-to-understand information provision, the highest and lowest values of the prior day are displayed along with and the current and lowest values of today. To check the amounts of solar radiation, wind speed, wind direction, and rainfall, a user must click a button, as shown in the figure presented above. When a user clicks a button, the weather information corresponding to the button is displayed. Figure 20b shows the webpage when the button “solar radiation” is clicked. In addition, the user can confirm the change in the weather data for one week by clicking the button “week” (Figure 20c). This webpage was evaluated highly by JA officials and agricultural workers. Moreover, in the provision of weather information, users can only check the data of their own WS.

Alert notifications at high and low temperatures to prevent damage to crops are described. Crops can be damaged severely at high or low temperatures. The alert notification is sent to farmers’ smartphones using LINE when the temperature is high or low. In the case of low temperatures, an alert notification is sent when the current temperature is below the set temperature. In the case of high temperatures, an alert notification is sent when the current base temperature is above the set temperature. The system will continue to send alert notifications until the user taps “Yes” twice. This means that the alert notification will stop when the user taps “Yes” twice. The alert notification is then disabled, and the alert notification is activated at 2:00 p.m. for low temperatures and at 5:00 a.m. for high temperatures.

Figure 21 presents an example of an alert notification at a high temperature. To stop the alert notification, a user must click “Yes” when the first notification arrives and click “Yes” again when the second notification arrives. Furthermore, if the user does not click, the user receives an alert notification periodically. Then, the notification stopped by the user is automatically resumed. The high temperature alert is at 2:00 a.m. and the low temperature alert is at 2:00 p.m.

The running cost of the cloud servers came to around USD 10 per WS each month. We received good feedback from farmers and JA staff that it was an appropriate fee.

## 7. Conclusions

This paper describes the development of the KOSEN-Weather Station to acquire stable weather information for a long time. Furthermore, we conducted basic experiments and field experiments. Then, a cloud server and a webpage for easy-to-understand weather information provision were produced. Alert notifications were sent to users’ smartphones to prevent damage to crops at high or low temperatures.

As a basic experiment, we compared meteorological data measured by KOSEN-WS and WXT520 and demonstrated that the weather station is fully available. In the field experiments, the KOSEN-WS’ stability and durability were examined when using a commercial power source and when using a solar panel and a battery. As shown by the experimental results, the KOSEN-WS was able to acquire meteorological data stably for up to 19 months or more and provide weather information to users. 

Moreover, results of the performance evaluation of KOSEN-WS’ durability demonstrated that the maximum temperature of operation is +44.9 °C and the minimum temperature is −16.5 °C. Weather information provided to agricultural workers was evaluated as easy to understand by agricultural workers and JA staff. Furthermore, alert notifications were given at high and low temperatures to prevent damage to crops because of temperature.

There were several points that could be considered in future work. First, there was a difference when comparing the measurement results of the KOSEN-WS with those of the reference device. In particular, there were differences in the measurement results for the wind speed and rainfall. To solve this problem, it is necessary to conduct measurements over a long period of time, clarify the causes, and make improvements. In addition, only the temperature was used for the alert notifications to farmers, but it is often necessary to make more complicated decisions based on multiple combined factors, such as the temperature, humidity, soil moisture, solar radiation, and the current level of growth. Next, if weather information in a local area can be collected stably over a long period of time, as in this system, it can be used in decision-making systems such as pesticide application timing and harvest time prediction, e.g., using the effective cumulated temperature. Furthermore, we will combine weather data with the know-how of skilled workers to schedule work efficiently using artificial intelligence to save labor for agricultural workers. A questionnaire survey of agricultural workers is necessary to improve the usability of the weather information provision. We plan to utilize the results of this survey to improve the usability of providing weather information to farmers. Finally, we plan to release the circuit diagram and source code of the KOSEN-WS and consider how to build a cloud server for providing weather information to farmers.

## Figures and Tables

**Figure 1 sensors-22-02108-f001:**
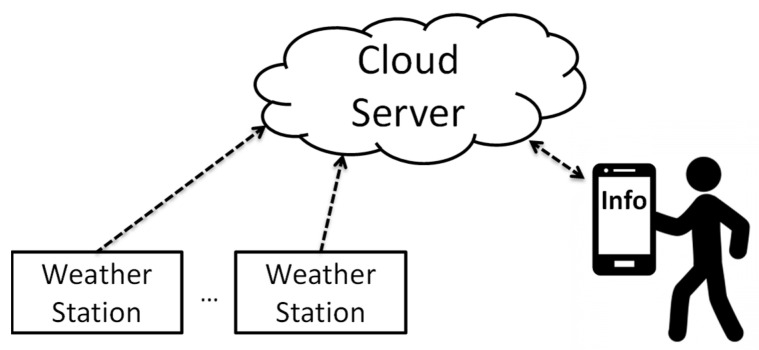
Overview of this system. The KOSEN-Weather Station transfers the weather data to a cloud server. Then, the cloud server accumulates meteorological data and provides agricultural workers with the necessary weather information.

**Figure 2 sensors-22-02108-f002:**
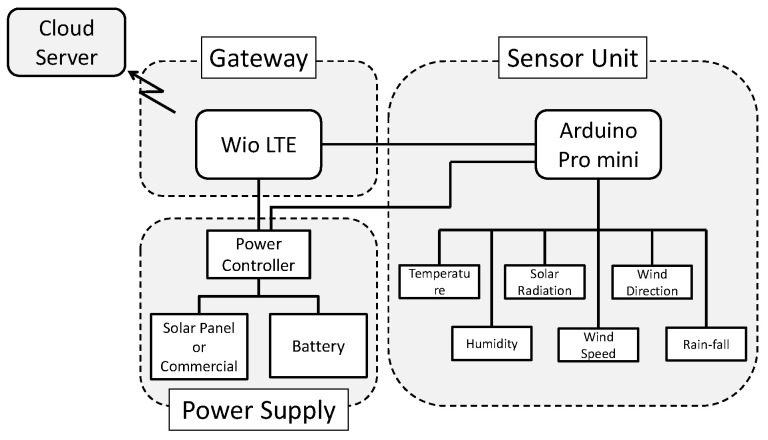
KOSEN-Weather Station configuration. KOSEN-WS consists of a sensor unit, a power supply, and a gateway.

**Figure 3 sensors-22-02108-f003:**
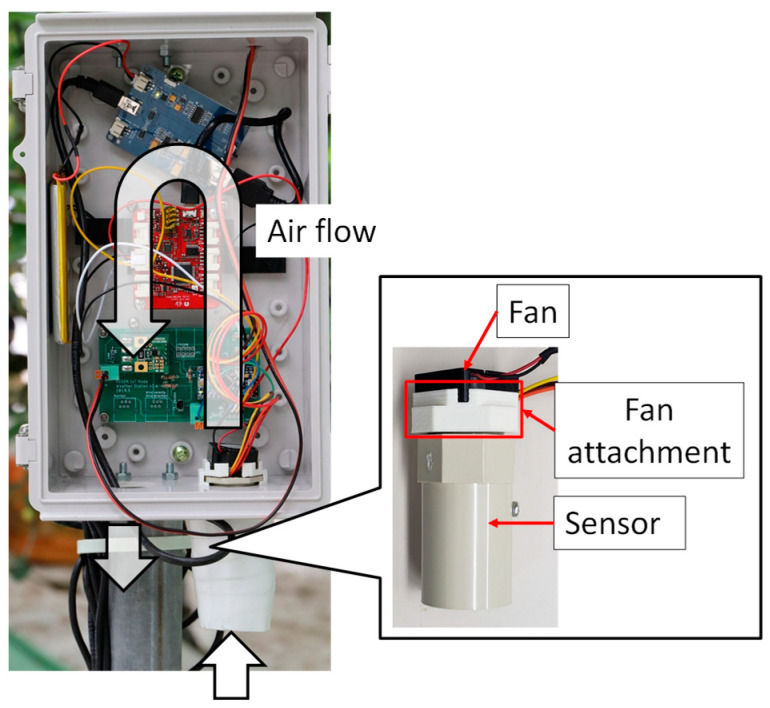
Temperature–humidity measuring device to which the simple Aßmann’s aspiration psychrometer is applied.

**Figure 4 sensors-22-02108-f004:**
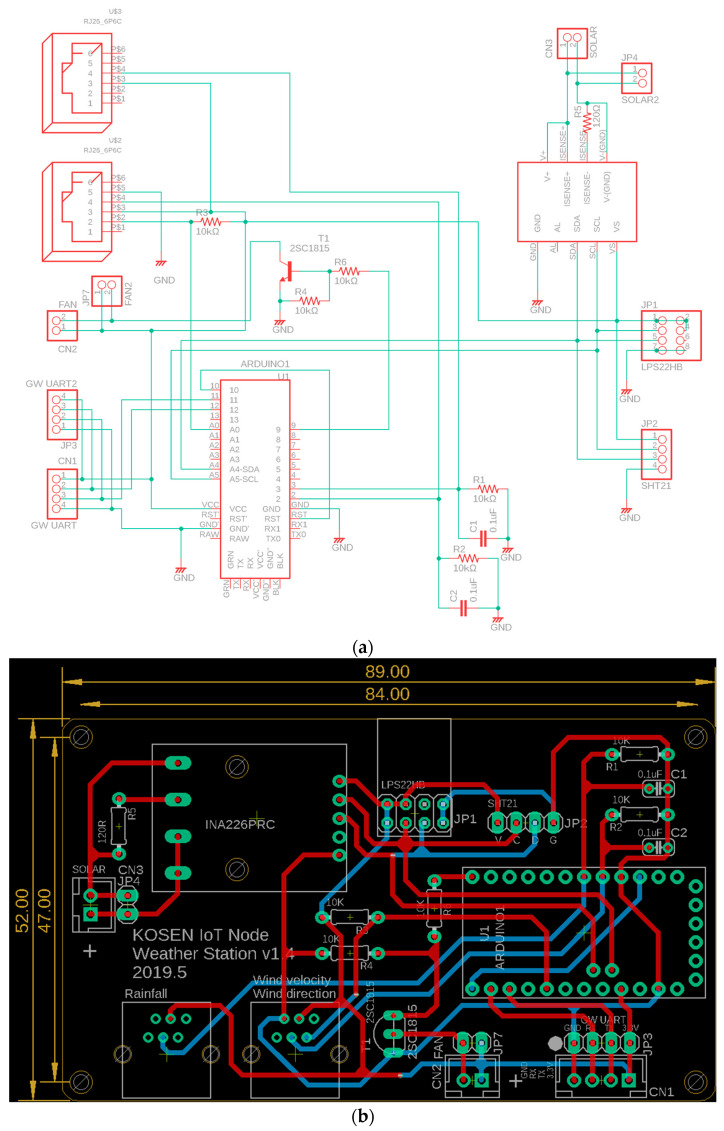
Circuit diagram and circuit board for the sensor unit. (**a**) Circuit diagram. (**b**) Circuit board.

**Figure 5 sensors-22-02108-f005:**
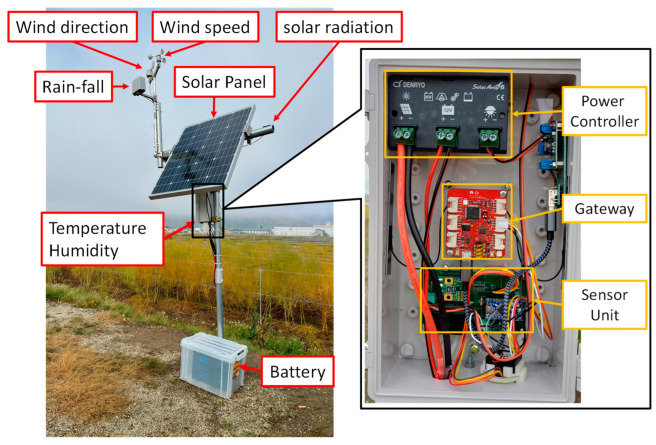
KOSEN-WS installed in a place where no power supply can be secured and its interior. The KOSEN-WS uses a power controller for solar panel and battery.

**Figure 6 sensors-22-02108-f006:**
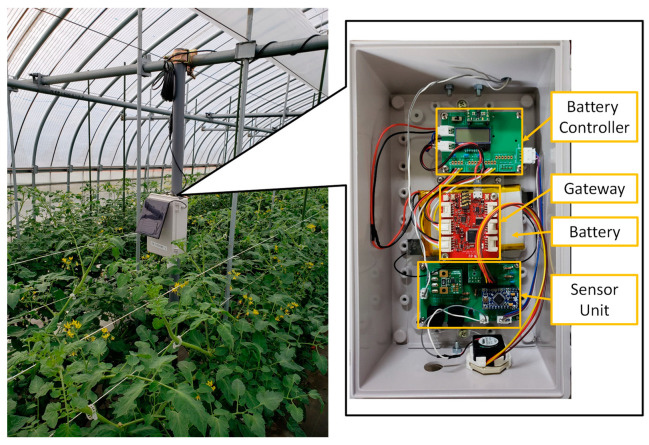
KOSEN-WS installed in a greenhouse with a commercial power supply and its interior. The KOSEN-WS was then installed at the growing point of the crop. The difference between the indoor KOSEN-WS and the outdoor KOSEN-WS (Figure 5) is that it uses a battery controller.

**Figure 7 sensors-22-02108-f007:**
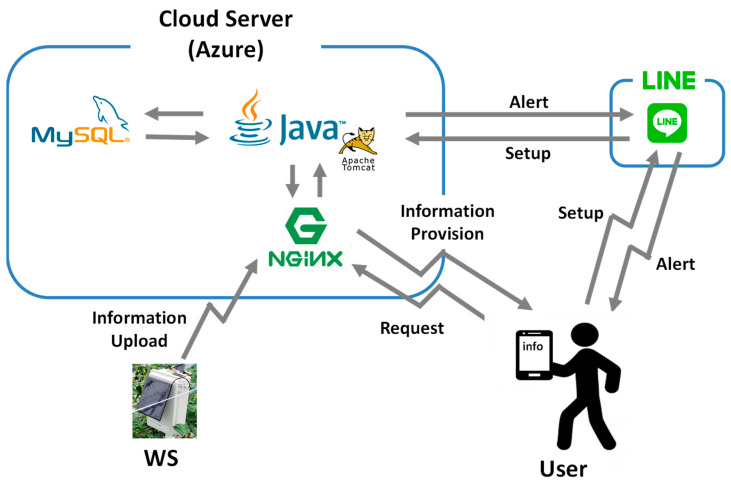
Overview of cloud server. KOSEN-WS forwards the measured data to the web server. The transmitted weather data are then stored in the database server via the application (AP) server. The user can check the weather information by accessing the webpage when necessary.

**Figure 8 sensors-22-02108-f008:**
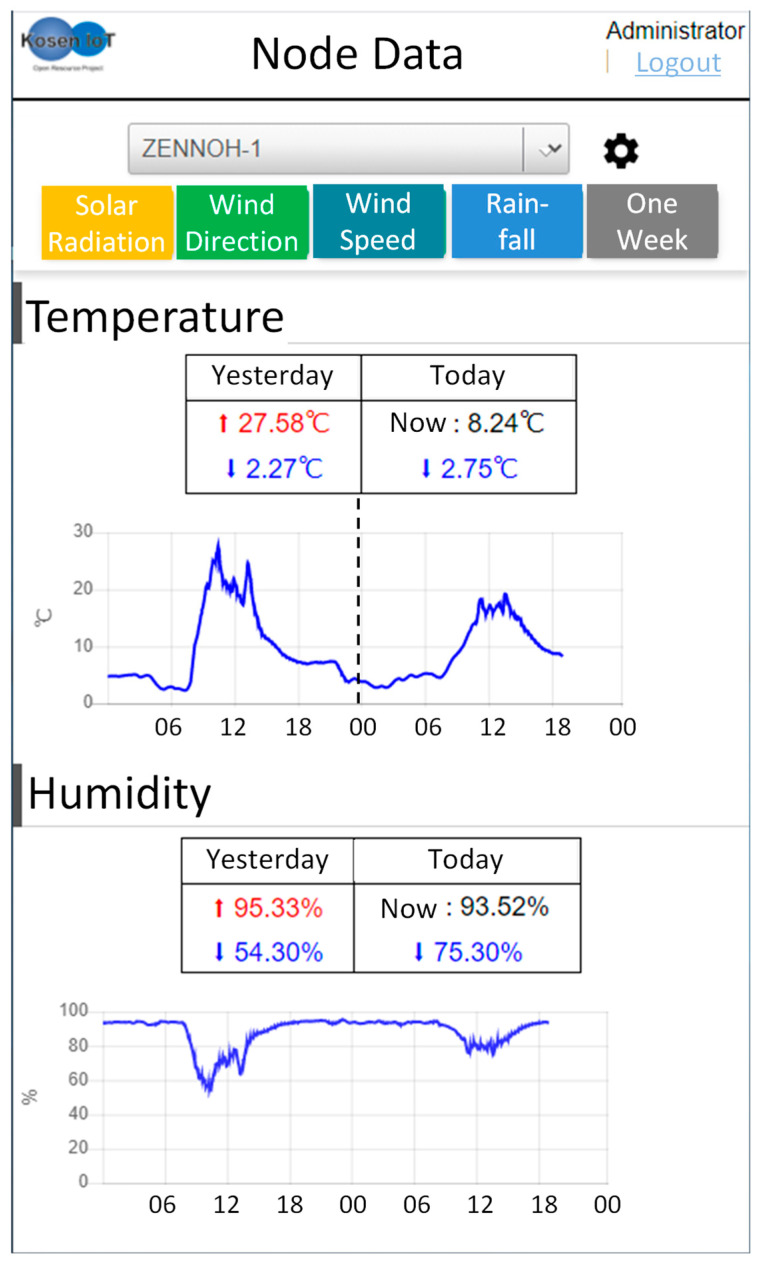
An example of a webpage for providing weather information to farmers. The default weather information to be displayed is temperature and humidity.

**Figure 9 sensors-22-02108-f009:**
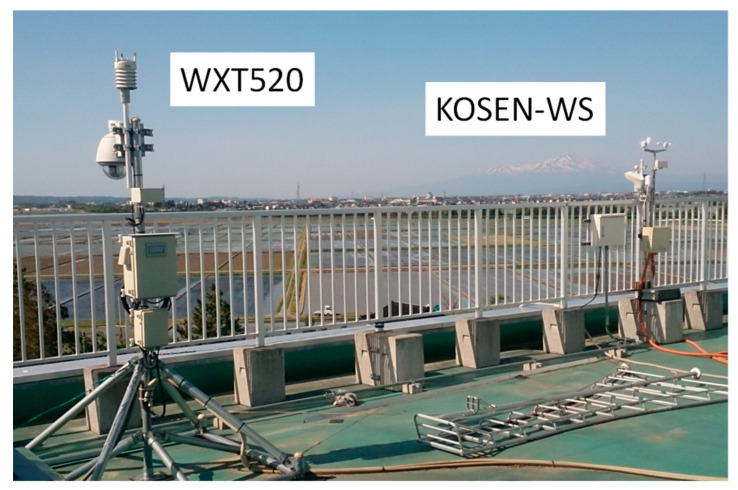
Installation locations of the KOSEN-WS and WXT520. The heights of the two airflow meters are the same, and the distance between the two devices is approximately 4 m.

**Figure 10 sensors-22-02108-f010:**
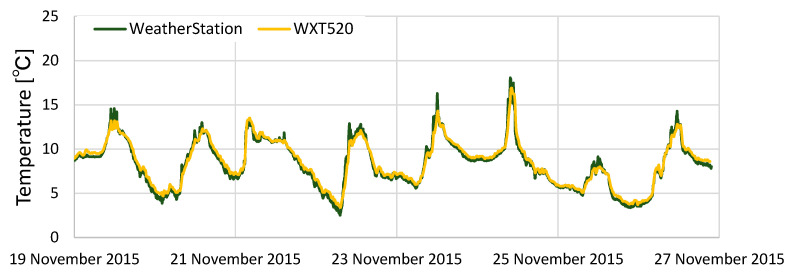
Results from comparing the temperature data of the KOSEN-WS and WXT520. The two sets of data are almost identical. These results show that the KOSEN-WS is effective for temperature/humidity measurements.

**Figure 11 sensors-22-02108-f011:**
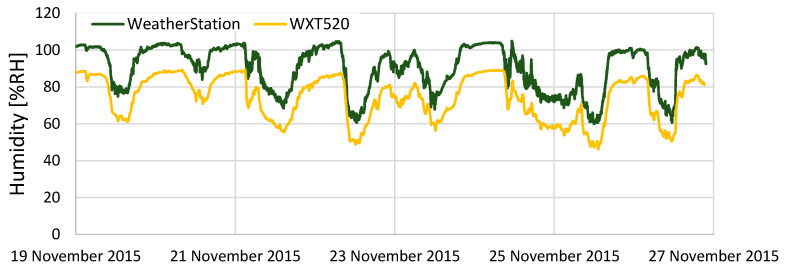
Results from comparing the humidity data of the KOSEN-WS and WXT520. It is difficult to use the humidity data because the difference between KOSEN-WS and the WXT520 is approximately 17% on average.

**Figure 12 sensors-22-02108-f012:**
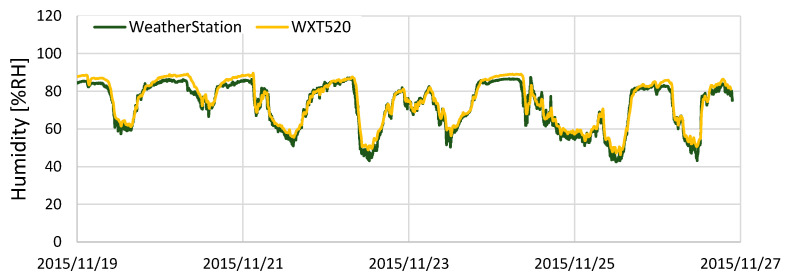
Results of comparing the humidity data after correction. Subtracting 17% from the humidity data measured by KOSEN-WS shows that the difference is smaller. This indicates that the corrected humidity data are valid.

**Figure 13 sensors-22-02108-f013:**
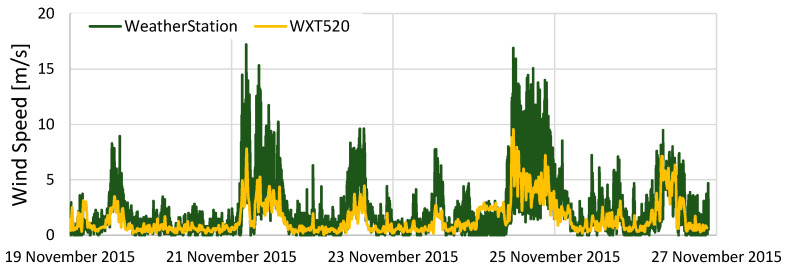
Results from comparing the wind speed of the KOSEN-WS and WXT520. The overall trend is the same, but there is a difference between the two measurement results. The KOSEN-WS measures using a wind vane and anemometer-type system, whereas the WXT520 measures using an ultrasonic-type system.

**Figure 14 sensors-22-02108-f014:**
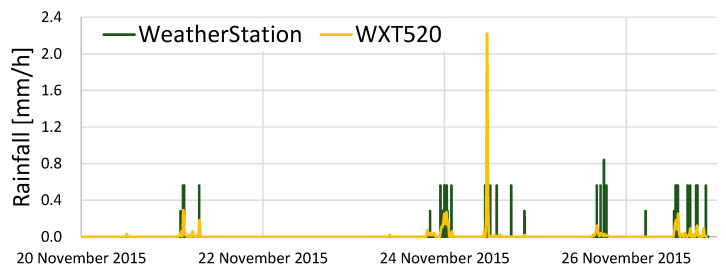
Results from comparing the rainfall data of the KOSEN-WS and WXT520. There is a large difference between the rainfall as measured by the KOSEN-WS and that measured by the WXT520. The KOSEN-WS uses an inexpensive tipping bucket rain gauge, which is not accurate.

**Figure 15 sensors-22-02108-f015:**
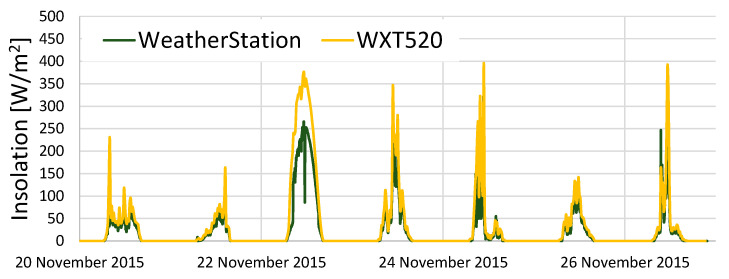
Results from comparing the solar radiation of the KOSEN-WS and WXT520. This shows that both sensors have almost the same results.

**Figure 16 sensors-22-02108-f016:**
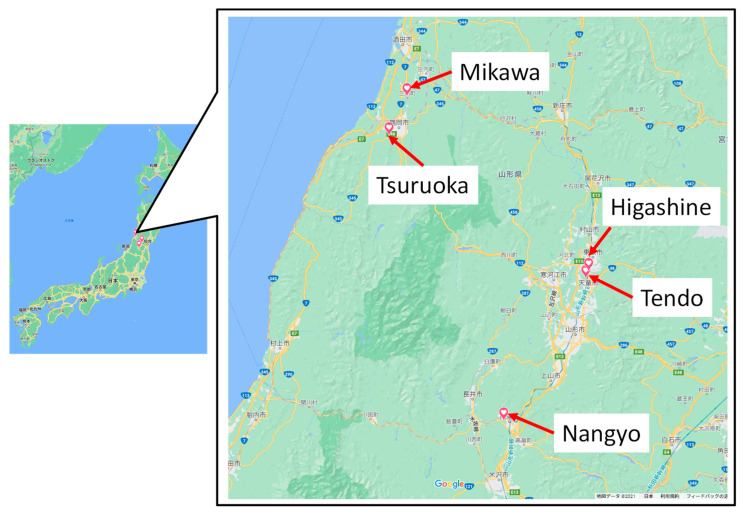
Installation places of KOSEN-WS in Yamagata Prefecture. Yamagata Prefecture is located approximately 300 km north of Tokyo. It is hot in summer and cold in winter. In winter, a large amount of solar radiation is low, owing to the large amount of snow.

**Figure 17 sensors-22-02108-f017:**
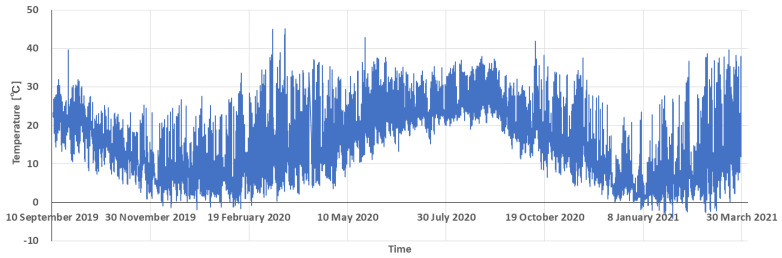
An example of the temperature data continuously measured for 19 months by the KOSEN-WS installed in Mikawa (greenhouse).

**Figure 18 sensors-22-02108-f018:**
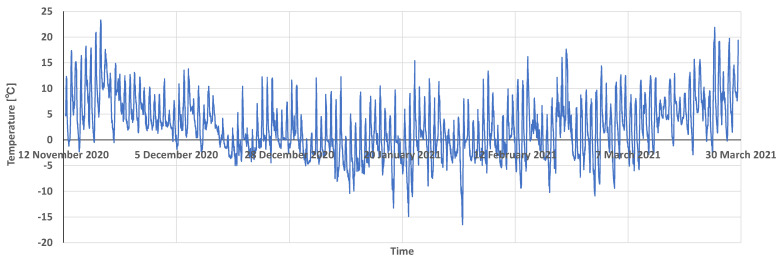
An example of temperature data measured by the KOSEN-WS installed in Nanyo.

**Figure 19 sensors-22-02108-f019:**
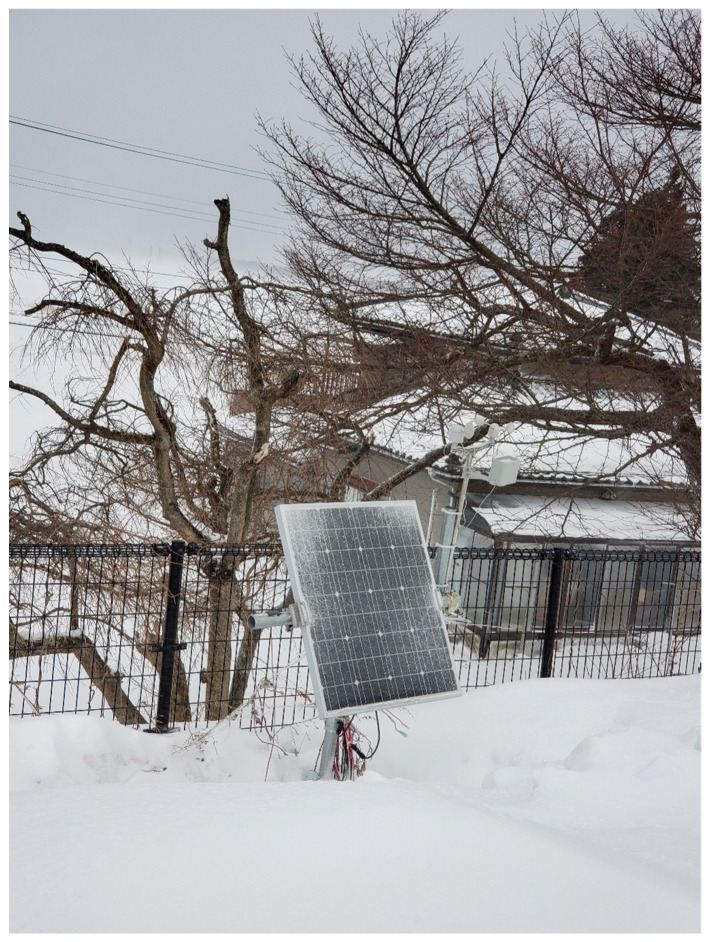
KOSEN-WS set up in Tsuruoka (outdoors) during winter. There is a significant amount of snow around the KOSEN-WS, but the KOSEN-WS is operating continuously.

**Figure 20 sensors-22-02108-f020:**
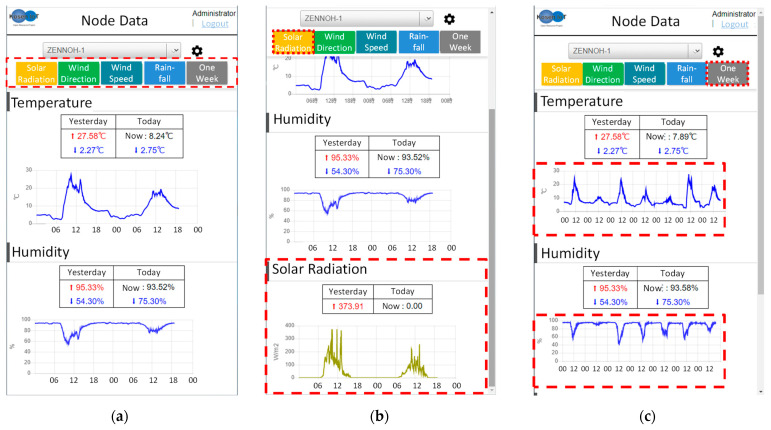
Webpage to provide weather information to farmers. (**a**) This figure is the default webpage. It shows the temperature and humidity, which are often checked by farmers. The highest and lowest values of the previous day and the current and lowest values of today are shown in numbers. To check other weather information, a user needs to click on a button (red dotted square), as shown in the figure. When a user clicks a button, the weather information corresponding to the button is displayed. To see the weather information of other KOSEN-WS, click on the blue dotted square and a list of KOSEN-WS will appear. (**b**) When clicking on the button of “solar radiation”, a graph of solar radiation will be added to the bottom of the webpage. In addition, other buttons can be clicked to add them to the bottom of the home page. Furthermore, when the user clicks on other buttons, weather information corresponding to the button will be added to the bottom of the webpage. (**c**) When clicking on the button of “one week”, this will present weather information for one week.

**Figure 21 sensors-22-02108-f021:**
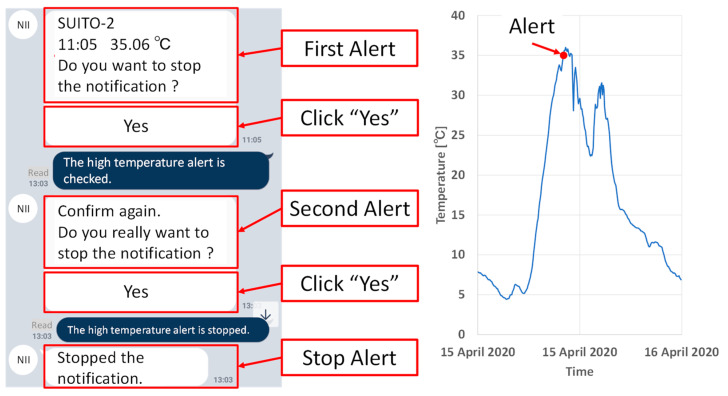
An example of alert notification at high temperature. The alert notification is sent to farmers’ smartphones using LINE when the temperature is over 35 °C. The figure on the right shows an alert notification by LINE. The user needs to click “Yes” after the first alert notification is received. After the second alert notification, the user needs to click “Yes” again. The user must click “Yes” twice to stop the alert notification. If the user does not confirm twice, the alert notification will continue to arrive.

**Table 1 sensors-22-02108-t001:** Conventional research on power-saving and inexpensive weather stations is summarized by communication method, measurement period, and sensors used.

Types of Microcomputer	Overview	Communication	Ref	Measurement Period	Sensors
Raspberry Pi	The Raspberry Pi plays the role of sensor measurement and gateway; the Raspberry Pi transmits the measured data to the server using 3G/4G, WiFi, Zigbee, LoRa, etc.	3G/4G	[13]	x	Gas, IR, moisture
			[14]	x	Temperature, wind, liquid level
		WiFi	[15]	x	Soil moisture, humidity, acoustic, etc.
			[16]	x	Temperature, accelerometer, etc.
			[17]	x	Camera
		Zigbee	[18]	x	Soil moisture
		LoRa	[19]	x	Temperature, humidity, etc.
Arduino	The Arduino plays the role of sensor measurement and gateway. The Arduino transmits the measured data to the server using 3G/4G, WiFi, Zigbee, LoRa, etc.	3G/4G	[20]	x	Soil moisture, temperature, humidity
			[21]	7 days	Temperature, soil temperature, rain, etc.
		WiFi	[22]	3 months	Temperature, humidity, pH, soil moisture, etc.
			[23]	x	Temperature, humidity, soil moisture, etc.
			[24]	Less than a day	Temperature, rain, etc.
		Zigbee	[25]	x	Temperature, soil moisture, rain
			[26]	2 days	Temperature, humidity, soil moisture
			[27]	x	Temperature, humidity, pH
		LoRa	[28]	7 days	Temperature, soil moisture, etc.
			[27]	x	Temperature, humidity, soil moisture, etc.
			[30]	1 day	Temperature, humidity, soil moisture, etc.
Raspberry Pi + Arduino	The Arduino plays the role of sensor measurement and the Raspberry Pi plays the role of gateway. The data measured by the Arduino are transferred to the cloud server via the Raspberry Pi.	3G/4G	[31]	Less than a day	Temperature, humidity, soil moisture, etc.
		WiFi	[32]	Less than a day	Temperature, humidity, etc.
		Zigbee	[33]	x	Temperature, soil moisture
		Wire	[34]	Less than a day	Temperature, soil moisture, etc.
Various micro-computers	Various microcomputers act as measurement sensors and gateways. The microcontrollers will transmit the measured data to the server via 3G/4G, WiFi, SigFox, etc.	3G/4G	[35]	x	Temperature, anemometer, etc.
			[36]	x	Soil moisture
		WiFi	[37]	Less than a day	Temperature, humidity
		SigFox	[38]	x	Temperature, humidity, etc.
		Bluetooth	[39]	1 day	Temperature, humidity, etc.

**Table 2 sensors-22-02108-t002:** Operating time and operating range of KOSEN-WS when using commercial power.

	Operating Time *			Temperature [°C]	
	Start	Stop	Continuous Operation	Maximum	Minimum
Mikawa (Indoor)	3 August 2019	Currently in operation	19 months	+44.9	−2.0
Mikawa (Indoor)	11 September 2019	Currently in operation	18 months	+44.8	−2.4
Mikawa (outdoor)	2 August 2019	26/05/2020	9 months	+43.6	−2.6
Higashine (Indoor)	27 May 2020	Currently in operation	10 months	+38.6	−10.9
Higashine (Indoor)	7 December 2019	Currently in operation	15 months	+38.0	−5.4

* The above measurement data were measured from August 2019 to March 2021.

**Table 3 sensors-22-02108-t003:** Operating time and operating range of KOSEN-WS when using solar panels and battery.

	Operating Time *			Temperature [℃]	
	Start	Stop	Continuous Operation	Maximum	Minimum
Tsuruoka (outdoor)	14 April 2020	18 October 2020	6 months	+41.3	−1.9
Tendo (outdoor)	6 October 2020	5 January 2021	3 months	+39.8	+1.1
Nanyo (Outdoor)	12 November 2020	currently in operation	4 months	+27.5	−16.5

* The above measurement data were measured from August 2019 to March 2021.

## Data Availability

The data that support the findings of this study are available from the corresponding author, Jeyeon Kim, upon reasonable request.

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
