# Peer review of "Development of KOSEN Weather Station and Provision of Weather Information to Farmers"

_sensors, 2022, doi:10.3390/s22062108_

Round 1
Reviewer 1 Report
Line 12 - Please introduce what you mean by "local areas."
Line 13 - "weather stations that are inexpensive, highly accurate, and have achieved stable measurements indoors and outdoors for long periods of time (over a year) are relatively rare." This idea implies that they exist. With the given advantages one wonders why these solutions are not more readily available. How does your work solve this issue?
Line 22 - "As a result, it is shown that KOSEN-WS is sufficiently available." Please review whether "available" is the intended word. Do you mean viable?
Line 26 - You provide an operational range from -16.5C to 44.9C, are these environmental or internal conditions? Have you measured the temperature of the chips?
Line 39 - Please introduce the ICT acronym.
Line 42 - The idea of the sentence is not clear, please review its redaction.
Line 75 - It is a widespread practice to generate fine-grained meteorological models from a coarse-grained input. We have the necessary information to generate accurate predictions based on well-known thermodynamics and fluid-mechanics models. However, these simulations are processing intensive. If the output is required within a small timeframe, or worse, on-demand then the models are simplified to reduce processing times. What would be more cost-effective, to improve the accuracy of the AMGSD or to get more sensors? Is it viable to distribute a sensor per square kilometer all over Japan? In the manuscript the 'simulation' notion is introduced as a drawback, however, the authors do not provide their reasonings for this assumption.
Line 103 - Review redaction.
Line 116 - Section 2 is Related work, but in Section 1 (Line 79) you also provide some related work. Some of the text is even repeated later. I recommend you merge the paragraph in Line 79 into Section 2.
Line 125 - It would be very useful to the reader if you tabulate some characteristics of current WS platforms: power requirements, physical size, cost, sensors, type of data collected. Please give concise numbers not just imply that the cost of something is "extremely high" or that it has a "high-power consumption." Table 1 is somewhat related, but Raspberry Pi and Arduino boards by themselves do not qualify in the same category of the WXT520.
Line 128 - "environ-mental"
Line 181 - "de-scribe"
Line 182 - Please reduce the usage of the word "However," it appears up to four times in a paragraph. Try alternatives like "Despite anything to the contrary"; "By contrast"; "On the other hand"; "All the same"; "At the same time"; "Be that as it may"; "Even so"; "Nevertheless"; "Nonetheless"; "Still"; and "Yet." The same goes for other terms like "In addition."
Line 312 - Are these costs in JPY?
Line 399 - I don't find convincing that the wind speed variation is caused by the difference of the sampling frequency. Did you try to match the WTX520 by sampling at 1/4s and comparing the results? Then there is a secondary explanation in line 405. Which is it?
Line 537 - The information on failures had been briefly provided in Line 500. I suggest creating a subsection containing all this information to prevent duplication.
Line 553 - Is the cost of this power controller included in the cost figures provided in Line 312?
Line 560 - There is an extra dot.
Line 603 - The description of the alert system could be improved. What happens if the high temperature or low temperature persists? Does the user keep getting notifications? If the user disables them, can they be enabled again through Line? The outdoor WS also issue notifications?
Line 650 - This paragraph is almost identical to the previous one.
* Overall, the proposed system intends to report temperature, humidity, solar radiation, wind speed, and rainfall. But for the latter two magnitudes, the measurements differ significantly from the expected results. It can tell that there is wind and that it is raining, but is such data useful? We can get as much information from the daily weather forecast on the news, or simply by looking out the window.
* Can any user with the URL consult all the data or only for their sensors? How do you manage the access rights to the platform? Please provide an overview of the security policies placed to protect the data.
* It would be particularly useful to provide a final comparison of the proposed system against the alternatives.
Author Response
Reviewer 1
Thank you for your suggestion. We have revised the manuscript as follows.
Line 12 - Please introduce what you mean by "local areas."
【Reply】
Local area in this paper indicates a small area such as a rice field, a greenhouse, etc. owned by an agricultural worker. The farmer then sets up this WS in the required local area and obtains weather information.
【Before correction】
There is an increasing demand for meteorological information, especially in local areas.
【After correction】Line 11-12
There is an increasing demand for meteorological information in local areas such as a rice field, a greenhouse, etc. owned by an agricultural worker.
Line 13 - "weather stations that are inexpensive, highly accurate, and have achieved stable measurements indoors and outdoors for long periods of time (over a year) are relatively rare." This idea implies that they exist. With the given advantages one wonders why these solutions are not more readily available. How does your work solve this issue?
【Reply】
The reason why the authors wrote "Relatively rare" is because there were no reports of WS satisfying these conditions as far as we could find.
And as mentioned in the manuscript, there are many studies on inexpensive WS. However, it is difficult to ensure durability and stability in fields during the summer because of the extremely high temperature and humidity. The authors developed the WS considering its durability and stability from the design stage.
The manuscript has been revised as follows.
【Before correction】
However, weather stations that are inexpensive, highly accurate, and have achieved stable measurements indoors and outdoors for long periods of time (over a year) are relatively rare.
【After correction】Line 13-15
However, weather stations that are inexpensive, highly accurate, and have achieved stable measurements indoors and outdoors for long periods of time (over a year) are not reported.
Line 22 - "As a result, it is shown that KOSEN-WS is sufficiently available." Please review whether "available" is the intended word. Do you mean viable?
【Reply】
The meaning of this sentence is that this WS is viable. The manuscript has been revised as follows.
【Before correction】
As a result, it is shown that KOSEN-WS is sufficiently available.
【After correction】Line 23
As a result, it is shown that KOSEN-WS is viable.
Line 26 - You provide an operational range from -16.5C to 44.9C, are these environmental or internal conditions? Have you measured the temperature of the chips?
【Reply】
Here, it means that KOSEN-WS was able to operate continuously under the environment of -16.5 ℃ to 44.9℃.
【Before correction】
Then, it was shown that the KOSEN-WS can operate from the maximum temperature of + 44.9 ℃ to the minimum temperature of -16.5 ℃.
【After correction】Line 26-27
Then, it was shown that the KOSEN-WS was able to operate continuously under the environment of -16.5 ℃ to 44.9℃.
【Addition】Line 651-652
That is, the KOSEN-WS was able to operate continuously under the environment of
-16.5 ℃ to 44.9℃.
Line 39 - Please introduce the ICT acronym.
【Reply】
As you indicated, the manuscript has been revised as follows.
【Before correction】
ICT in agriculture
【After correction】Line 39-40
Information and Communication Technology (ICT) in agriculture
Line 42 - The idea of the sentence is not clear, please review its redaction.
【Reply】
As you indicated, we have revised the following.
【Before correction】
Smart agriculture is a new type of agriculture in which robotics and ICT are used to promote such things as labor saving, precision, and high-quality production.
【After correction】Line 42-44
Smart agriculture refers to the use of technologies such as robotics and ICT on farms to achieve labor saving, high-quality production, and so on.
- Line 75 - It is a widespread practice to generate fine-grained meteorological models from a coarse-grained input. We have the necessary information to generate accurate predictions based on well-known thermodynamics and fluid-mechanics models. However, these simulations are processing intensive. If the output is required within a small timeframe, or worse, on-demand then the models are simplified to reduce processing times. What would be more cost-effective, to improve the accuracy of the AMGSD or to get more sensors? Is it viable to distribute a sensor per square kilometer all over Japan? In the manuscript the 'simulation' notion is introduced as a drawback, however, the authors do not provide their reasonings for this assumption.
【Reply】
The KOSEN-WS described in this paper is not supposed to set up WS all over Japan. This WS is aimed to be installed only in the local area where farmers need it and to provide weather information at that place.
While simulations can provide detailed meteorological information, existing meteorological observation equipment is currently incapable of predicting gusty winds, tornadoes, guerrilla rain, unexpectedly strong rain, sudden rain, etc. in the local area. And since the authors are considering the provision of detailed weather information in a local area, the authors believe that the current simulation technology is not capable of achieving the accuracy required by farmers. We will refrain from answering this question.
【Addition】Line 78-80
These systems are unable to precisely predict gusty winds, tornadoes, guerrilla rains, unexpectedly strong rains, sudden rains, etc. in the local area at present.
Line 103 - Review redaction.
【Reply】
We wrote this sentence in the meaning of "The WS that can be measured stably over a long period of time is necessary," but the translation is wrong.
【Before correction】
The WS that can be measured weather information stably for a long time is studied.
【After correction】Line 92
The WS that can be measured stably over a long period of time is necessary.
Line 116 - Section 2 is Related work, but in Section 1 (Line 79) you also provide some related work. Some of the text is even repeated later. I recommend you merge the paragraph in Line 79 into Section 2.
【Reply】
As you pointed out, I moved the duplicate contents to 2. Related work.
【After correction】Line 81-84
Various studies on the acquisition of meteorological information in the local area have been actively conducted to solve these problems [7]-[40]. Nevertheless, there has been insufficient research on WSs that are inexpensive and can acquire weather information stably over a long period of time. The details are described in Section 2.
【After correction】Line 111-115
The WXT520 Series Weather Transmitter [7] is an integrated meteorological instrument that can observe meteorological data of six kinds. It is possible to connect an external sensor to configure a meteorological observation hub. The WXT520 can acquire weather information stably. However, it is necessary to use an AC power supply because of its power consumption and the price is high.
10.1
Line 125 - It would be very useful to the reader if you tabulate some characteristics of current WS platforms: power requirements, physical size, cost, sensors, type of data collected. Please give concise numbers not just imply that the cost of something is "extremely high" or that it has a "high-power consumption."
【Reply】
The specifications of KOSEN-WS that can be used by small-scale farmers are summarized as follows based on the questionnaire survey by farmers and JA staff, and the author's knowledge.
【Addition】Line 181-191
Based on the above, the specifications of KOSEN-WS that can be used by small-scale farmers are summarized as follows based on the questionnaire survey by farmers and JA staff, and the author's knowledge. The size of the WS should be less than 200mm in width, less than 300mm in height, and less than 150mm in depth because the size of the WS should be small enough for farmers to install easily. The cost of the WS should be affordable for small scale farmers, so the price should be less than $1000, and the monthly maintenance fee should be less than $10. It is necessary to operate continuously for more than one year. Finally, the information to be collected includes temperature, humidity, solar radiation, rainfall, wind speed, and rainfall, which are all necessary for agriculture. In addition, the device should have a structure that allows farmers to install only the sensors they need.
10.2
Table 1 is somewhat related, but Raspberry Pi and Arduino boards by themselves do not qualify in the same category of the WXT520.
【Reply】
The WS presented in Table 1 are relatively inexpensive. The Raspberry Pi and Arduino are the types of microcomputers used in the WSs. This is not for comparison with the WXT520. We have added an item to Table 1.
Line 128 - "environ-mental"
Line 181 - "de-scribe"
【Reply】
As you pointed out, the above mistakes have been corrected throughout this manuscript.
Line 182 - Please reduce the usage of the word "However," it appears up to four times in a paragraph. Try alternatives like "Despite anything to the contrary"; "By contrast"; "On the other hand"; "All the same"; "At the same time"; "Be that as it may"; "Even so"; "Nevertheless"; "Nonetheless"; "Still"; and "Yet." The same goes for other terms like "In addition."
【Reply】
As you indicated, we have made revisions throughout this paper.
13.
Line 312 - Are these costs in JPY?
【Reply】
The Japanese yen is converted to dollars for representation. One dollar is about 110 yen.
Line 399 - I don't find convincing that the wind speed variation is caused by the difference of the sampling frequency. Did you try to match the WTX520 by sampling at 1/4s and comparing the results? Then there is a secondary explanation in line 405. Which is it?
【Reply】
The anemometer (Wind / Rain Sensor Assembly [42]) used in this WS is an inexpensive one, and its precision is not good. The cause of the error is due to the measurement method. The measurement method of this anemometer converts the wind speed using the number of rotations per second. In other words, sampling of less than 1 second is not possible. Because of these factors, measurement precision is poor. Therefore, the measurement precision of this anemometer is not good. Further, this anemometer cannot be compared to the WXT520 because it cannot sample 1/4s.
And the line 405 was deleted because it is redundant.
【Before correction】
Figure 13 shows the results of the comparison of wind speeds. The overall trend is the same, but there is an error between the two measurement results. The cause of the error is the difference in measurement method. The KOSEN-WS measures wind speed every second and outputs the average of 5 seconds, while the WXT520 measures wind speed every 1/4s by default.
【After correction】Line 561-567
Figure 13 shows the results of the comparison of wind speeds. The overall trend is the same, but there are errors between the two measurement results. The measurement method of this anemometer converts the wind speed using the number of rotations per second. In other words, sampling of less than 1 second is not possible. The KOSEN-WS measures wind speed every second and outputs the average of 5 seconds, while the WXT520 measures wind speed every 1/4s by default. Because of these factors, the error between the two anemometers occurred.
15.
Line 537 - The information on failures had been briefly provided in Line 500. I suggest creating a subsection containing all this information to prevent duplication.
【Reply】
As you suggested, these contents have been moved to Discussion.
【After correction】Line 679-693
Next, the causes of the WS failure are described. At the beginning of the installation, it sometimes stopped in 2 weeks or 2 months. The cause of the outage was both software and hardware. Causes of program stoppage included counter overflow and reset timing error. There were also cases when the microcontroller stopped because it could not take measurements due to a sensor failure. Counter overflow and reset timing errors were corrected by reviewing the program and correcting the counter. And as a countermeasure for sensor failures, the microcontroller was able to operate continuously without stopping by returning NaN when the sensor could not be measured. Causes of failure in hardware included component mounting error due to insufficient solder, short-circuit due to overcurrent, and failures due to corrosion. In case of failure due to short-circuit caused by overcurrent or corrosion, we redesigned the circuit board. Furthermore, in order to use the KOSEN-WS for a long period of time, the circuit board of the KOSEN-WS needs to be converted to printed circuit board (PCB) assembly. Currently, KOSEN-WS is easy to be corroded by moisture because we soldered the components. But it is possible to prevent corrosion due to moisture by PCB assembly of the KOSEN-WS.
Line 553 - Is the cost of this power controller included in the cost figures provided in Line 312?
【Reply】
Yes, the WS price includes the price of the Power controller.
17.
Line 560 - There is an extra dot.
【Reply】
As you mentioned, I revised the manuscript. And I corrected the typos and other errors throughout the whole manuscript.
18.
Line 603 - The description of the alert system could be improved. What happens if the high temperature or low temperature persists? Does the user keep getting notifications? If the user disables them, can they be enabled again through Line? The outdoor WS also issue notifications?
【Reply】
Alert notifications are sent to users as follows. In the case of low temperature, an alert notification is sent when the current temperature is below the set temperature. In the case of high temperatures, an alert notification is sent when the current base temperature is above the set temperature. The system will continue to send alert notifications until the user taps Yes twice. This means that the alert notification will stop when the user taps Yes twice. The alert notification is then disabled, and the alert notification is activated at 2:00 p.m. for low temperature and at 5:00 a.m. for high temperature. Currently, if alert notification is disabled, it cannot be enabled by the user. The reason for this is due to the opinions of farmers and JA staff. We decided that resetting the system once a day is sufficient unless there is an abnormal weather condition. We will make it enabled if there are more opinions from farmers that it is necessary.
This alert system will provide alert notifications for both indoor and outdoor use as well.
【Before correction】
Alert notifications at high and low temperatures to prevent damage to crops are described. Crops can be damaged severely at high or low temperatures. For example, if the temperature inside the greenhouse is 35 °C or higher, then it is necessary to open the greenhouse window to lower the temperature. The alert notification is sent to farmers' smartphones using LINE when the temperature is high or low. In this paper, the alert notification is sent when the temperature exceeds the set temperature that is 35 °C in this paper.
【After correction】Line 746-755
Alert notifications at high and low temperatures to prevent damage to crops are described. Crops can be damaged severely at high or low temperatures. The alert notification is sent to farmers' smartphones using LINE when the temperature is high or low. In the case of low temperature, an alert notification is sent when the current temperature is below the set temperature. In the case of high temperatures, an alert notification is sent when the current base temperature is above the set temperature. The system will continue to send alert notifications until the user taps “Yes” twice. This means that the alert notification will stop when the user taps “Yes” twice. The alert notification is then disabled, and the alert notification is activated at 2:00 p.m. for low temperature and at 5:00 a.m. for high temperature.
19.
Line 650 - This paragraph is almost identical to the previous one.
【Reply】
We have included a paragraph that should be deleted. As you indicated, the manuscript has been revised as follows.
【After correction】Line 787-804
There were several points that could be considered in future work. First, there was a difference when comparing the measurement results of the KOSEN-WS with those of the reference device. In particular, there were differences in the measurement results for the wind speed and rainfall. To solve this problem, it is necessary to conduct measurements over a long period of time, clarify the causes, and make improvements. In addition, the only temperature was used for the alert notifications to farmers, but it is often necessary to make more complicated decisions based on multiple combined factors, such as the temperature, humidity, soil moisture, solar radiation, and the current level of growth. Next, if weather information in a local area can be collected stably over a long period of time, as in this system, it can be used in decision-making systems such as pesticide application timing and harvest time prediction, e.g., using the effective cumulated temperature. Furthermore, we will combine weather data with the know-how of skilled workers to schedule work efficiently using artificial intelligence to save labor for agricultural workers. A questionnaire survey of agricultural workers is necessary to improve the usability of the weather information provision. We plan to reflect the results of this survey to improve the usability of providing weather information to farmers. Finally, we plan to release the circuit diagram and source code of the KOSEN-WS and how to build a cloud server for providing weather information to farmers.
20.
* Overall, the proposed system intends to report temperature, humidity, solar radiation, wind speed, and rainfall. But for the latter two magnitudes, the measurements differ significantly from the expected results. It can tell that there is wind and that it is raining, but is such data useful? We can get as much information from the daily weather forecast on the news, or simply by looking out the window.
【Reply】
As you pointed out, the wind speed and rainfall gauges used in this study are inexpensive and insufficiently accurate. Especially in the case of rainfall, there is a difference from the reference, and it may not be worth using.
However, the reason for setting up a WS in a local area is that weather information obtained from a wide region representative location like AMEDAS is often different from that of the user's rice field or greenhouse. In other words, it is necessary to have weather information for the location that the user needs. As mentioned in question 7, wind gusts, tornadoes, guerrilla rains, unexpectedly strong rains, sudden rains, etc. in specific areas cannot be predicted with current observation instruments. In the case of rain gauges, improvements are needed, but it is possible to check graphs on days when rainfall is high. We also believe that anemometers can be used for this purpose. This is especially effective for greenhouses.
This system can be connected to a maximum of six sensors, and if necessary, only the necessary sensors can be installed. And if the anemometer and rain gauge are not needed, farmers can be not installed. As mentioned in the future issues, the anemometer, wind vane, and rain gauge need to be improved.
21.
* Can any user with the URL consult all the data or only for their sensors? How do you manage the access rights to the platform? Please provide an overview of the security policies placed to protect the data.
【Reply】
Administrators (authors) can check all the data. And users can check the data only for their own WSs. Since this research is conducted in collaboration with JA, we have an agreement on these matters, and they are properly managed.
【Addition】Line 733-734
And in the provision of weather information, users can only check the data of their own WSs.
22.
* It would be particularly useful to provide a final comparison of the proposed system against the alternatives.
【Reply】
We compared the conventional WS with the proposed WS in terms of price, precision, operating time, and ease of installation. In conventional research, inexpensive WS and precise WS have already existed respectively. However, there is no WS that satisfies all the requirements such as price, precision, operating time, and ease of installation except for this KOSEN-WS.
【Addition】Line 657-661
We compared the conventional WS with the proposed WS in terms of price, precision, operating time, and easy-to-install. In conventional research, inexpensive WS, precise WS, and easy-to-install WS have already existed respectively. However, there is no WS that satisfies all the requirements such as price, precision, operating time of more than one year, and easy-to-install except for this KOSEN-WS.

Reviewer 2 Report
This manuscript presents the development of inexpensive, highly accurate, and durable weather stations for farmers in Japan. At a glance, the technical construction of the device seems appropriately conducted. However, some missing information and inadequate presentations require significant revisions/additions. The following comments intend to enable the authors to disseminate their work at the highest possible quality.
- The problem of engineering design.
- After reading this first version, the first curiosity is o the lack of a formal process for designing the KOSEN Weather Station device. While I believe that the authors are experts in relevant fields, scientific technology design should undergo within the corridor of a formal engineering design of the technology. Without a solid basis on an established engineering design methodology, the scientific process in designing the technology appears arbitrary and sometimes one-sided/armchaired. The authors must add a new separate section to explain their engineering design methodology (for example, using VDI 2222 or any other established engineering design) to convince readers that their technology is the product of a systematically managed design process.
- The problem of affordability.
- The authors repeatedly claim that the device being developed (and some parts of it) is "inexpensive." However, it is unclear to whom the device is inexpensive. Being inexpensive does not mean that the device will be affordable to its targeted market. Therefore, the authors must carefully explain their targeted market, the parameters for the affordability (investment and cost for buying, repairing, maintaining, reselling, etc.), and how the investment & cost of the device match with the willingness to invest/spend of the targeted market.
- The problem of generalized durability.
- The authors claim that the KOSEN WS device is durable and capable of long-term use (over a year). However, it is unclear whether long-term use is even necessary for contextual outdoor use in Japan. Common crops in Japan are grown outside the winter season, while those grown in the winter season are planted inside a maintained/protected environment. Therefore, the generalization of durability for both outdoor and indoor is probably an over-specification, which would make it inefficient for the cost to operate it outdoor throughout the year. Therefore, the authors should reconsider their generalized claim of durability and avoid unnecessary outdoor performance that merely adds running costs without being useful for the whole year.
- The problem of data gathering on opinions.
- Lines 563-565. The authors mention "the opinions of agricultural workers and Japan Agricultural Cooperatives Zen-Noh Yamagata (JA) staff" to guide the development of User Interface (UI) for potential recipients of information from the KOSEN device. However, there is no explanation of how the opinions were gathered, including the instrument to gather the opinions, the validation of the opinions, and the synthesis of the opinions. It raises doubts whether those "opinions" are scientifically justified. The authors must add a specific subsection to explain the scientific process of gathering the "opinions," which includes the approach, instrument, validation, and synthesis/analysis process. This section could be added into the new "Methodology" section explaining the engineering design methodology.
Author Response
Reviewer 2
Thank you for your suggestion. We have revised the manuscript as follows.
- The problem of engineering design.
After reading this first version, the first curiosity is o the lack of a formal process for designing the KOSEN Weather Station device. While I believe that the authors are experts in relevant fields, scientific technology design should undergo within the corridor of a formal engineering design of the technology. Without a solid basis on an established engineering design methodology, the scientific process in designing the technology appears arbitrary and sometimes one-sided/armchaired. The authors must add a new separate section to explain their engineering design methodology (for example, using VDI 2222 or any other established engineering design) to convince readers that their technology is the product of a systematically managed design process.
【Reply】
As you mentioned, we describe the methodology of engineering design in the design of WS. We explain in the following order: problem definition, opinion, instrument, validation, and synthesis/analysis.
【Addition】Line 347-464
3.4 Methodology
We describe the methodology of engineering design in the design of WS and provision of weather information to farmers in the following order: problem definition, opinion, instrument, validation, and synthesis/analysis.
3.4.1 Weather Station
・ Problem; Weather information can be applied to understanding growth conditions, predicting harvest time, countermeasures against pests and diseases, climate change and abnormal weather, and predicting freezing and frost damage. It is important to obtain stable weather information over a long period. However, there is a lack of studies on WSs that are inexpensive, precise, and capable of obtaining stable weather information over a long period in local areas.
・ Opinions; To obtain an understanding of the current situation in the agricultural field and to collect the various opinions of farmers, we conducted a literature review and a questionnaire survey of farmers and JA staff. We conducted a questionnaire survey on the types of weather information needed for agricultural work and what kind of weather information should be provided to users. The questionnaire items include the type of crop, the type of weather information required (sensor type), the period of use, the price, the size, and the installation location. Based on the results of the questionnaire survey and the knowledge of the authors, the specifications to implement the WS were determined.
・ Instrument; The results of the questionnaire survey and the authors’ knowledge are summarized to develop a WS that can acquire weather information stably over a long time. We describe sensor, microcomputer, communication device, circuit board design, power supply, power consumption, size of WS, and server construction. We surveyed weather information that is necessary for agricultural work. Then, we summarized the measurement precision, price, and measurement range of sensors for each weather information.
We examine whether the microcomputer can measure various sensors easily, whether the microcomputer can operate stably for a long time, and so on. To ensure stable transmission of meteorological information to the server at any location, the communication device was investigated. The following will be discussed. It is necessary to consider the connection method between the microcomputer and the communication device. In general, the most common method is to connect to the microcomputer using a USB connection or a shield. Some microcomputers have a built-in communication function. It is necessary to consider the compatibility between the communication device and microcomputer, and corrosion in a high temperature and humidity environment. Microcomputers with built-in communication functions are more expensive, but they can operate stably and continuously and can be used for a long time even in hot and humid environments. We discussed communication systems. The most used communication systems are Cellular networks and Low Power Wide area (LPWA). Cellular networks (4G and 3G) have a wide communication range and small initial costs. However, Communication devices using cellular networks consume a lot of power and have running costs.
LPWA (Low Power Wide Area) includes ZigBee, LoRa, Sigfox, and so on. Communication devices using LPWA have low power consumption and low running cost. However, LPWA may cause communication instability due to crop growth. In addition, the initial cost is high due to the need for a repeater nearby to transmit weather information to the server.
We will discuss the circuit board design. It is necessary to design a durable circuit board that can be used for a long time in indoor and outdoor environments. Printed circuit board (PCB) and printed circuit board assembly (PCBA) are recommended for stable measurement over a long time. PCBs are inexpensive, but require soldering, require countermeasures against corrosion, etc. PCBAs are more expensive, but do not require soldering and are more durable.
We will discuss power consumption. To use the WS for a long time even outdoors, it is better to have low power consumption. In general, the power consumption of microcomputers is highest during communication. It is necessary to supply sufficient power even when using the cellular network.
It is necessary to consider where power is secured and where it is not. In places where power is secured, we can use a USB adapter. However, in places where power is not secured, it is necessary to use solar panels and batteries in consideration of the power consumption of a WS. Since Japan has a rainy season, it is necessary to secure a power source that can be used during this period. The amount of power generated by the solar panel and the battery capacity should be considered.
The WS should need to be of a size that is easy for farmers to install, and the WS should not interfere with farm work after installation.
We will discuss items to be considered when building a server. It is necessary to build a server to accumulate weather information uploaded from WS and to provide weather information to users. There are two types of servers: on-premises and cloud servers. On-premises have a high initial cost, but low running cost. And it is more secure in terms of security but requires more expertise. Cloud servers have low initial costs and are highly scalable. However, running costs are high.
A prototype of the WS was produced by summarizing the above information. The microcomputer for sensor measurement is an Arduino Pro Mini. And since many fields do not have access to the Internet, we chose WioLTE, a microcomputer with a communication system, for stable uploading of weather data to the server. It is also easy to program because it is interchangeable with the commonly used Arduino. For the circuit board design, we decided to use PCBs, because the cost is low, although PCB soldering is necessary. The WS was made to be easy for users to install, considering the opinions of agricultural workers. The size of the box to hold the microcomputer, various sensors, and power supply controller was less than 300 x 200 x 150 mm, and the box was made easy to fix using clamps. For the power source, we selected a solar panel and a battery with a capacity of 100W and 12V72Ah, respectively, to ensure power supply even during the rainy season in Japan. The solar panel has a power generation capacity of 100W, and the battery has a capacity of 12V72Ah. 25W and 12V30Ah were tested, but the amount of solar radiation and the battery capacity was not enough for continuous operation during the rainy season.
We decided to use a cloud server. Cloud servers require running costs, but the initial cost is small and maintenance is simple.
・ Validation; In the basic experiment, we validated whether this prototype can be used in a real environment. The WS and the reference WS were installed at the same place, and the precision of the sensors was compared. In case of WS failure, the causes were listed and countermeasures were taken.
We solved the problems that occurred in the basic experiment and conducted the field experiment, which is a hot and humid environment. In the field experiment, we investigated whether the WS can operate stably and continuously for a long time in the real environment. We conducted field experiments in four places (fields and greenhouses with power supply, and fields and greenhouses without power supply) to verify whether the WS can operate stably and continuously for a long time. Then, in case of WS failure, the causes were listed and countermeasures were taken.
We solved the above problems by repeating the same process as [instrument] and [validation] again until all the problems are cleared.
・ Synthesis/Analysis; In case of WS failure, we retrieved them, analyzed the causes of failure from both hardware and software aspects, and solved the problems.
In the basic experiment, we carried out the operation check of the prototype and analyze the hardware and software problems. For hardware issues, we analyzed the precision of the sensor, the soldering of the board, and the capacity of the solar panel and battery in places where power cannot be secured, and discussed solutions. For software issues, we analyzed the microcontroller program, sensor measurement by the microcontroller, and whether or not the measurement data can be uploaded to a server, and discussed solutions.
In the field experiment, we analyzed hardware and software problems to verify that the WS could operate stably and continuously for a long time. Hardware problems included failures of each sensor, short-circuit generation due to over-current, and corrosion and peel-off of soldering in a hot and humid environment. Software problems included the occurrence of overflow and microcontroller shutdown due to sensor failure. The solutions to these problems were discussed.
The above problems were solved by repeating the same process as [instrument] and [validation].
- The problem of affordability.
The authors repeatedly claim that the device being developed (and some parts of it) is "inexpensive." However, it is unclear to whom the device is inexpensive. Being inexpensive does not mean that the device will be affordable to its targeted market. Therefore, the authors must carefully explain their targeted market, the parameters for the affordability (investment and cost for buying, repairing, maintaining, reselling, etc.), and how the investment & cost of the device match with the willingness to invest/spend of the targeted market.
【Reply】
Our target market is small- and medium-scale farmers. There are about 1 million small- and medium-scale farmers in Japan. The standard for low price is $ 1000. The reason is a result of discussions with JA staff, with whom we are currently conducting joint research.
And we believe that "how the investment & cost of the device match with the willingness to invest/spend of the targeted market." is a perfect match. We are currently in discussions with JA staff about the commercialization and sale of WS, but cannot give details about these.
- The problem of generalized durability.
The authors claim that the KOSEN-WS device is durable and capable of long-term use (over a year). However, it is unclear whether long-term use is even necessary for contextual outdoor use in Japan. Common crops in Japan are grown outside the winter season, while those grown in the winter season are planted inside a maintained/protected environment. Therefore, the generalization of durability for both outdoor and indoor is probably an over-specification, which would make it inefficient for the cost to operate it outdoor throughout the year. Therefore, the authors should reconsider their generalized claim of durability and avoid unnecessary outdoor performance that merely adds running costs without being useful for the whole year.
【Reply】
In the case of outdoor crops, the operating time of WS may be sufficient for about 7 months out of the year. However, some outdoor crops are double-cropping or double-season crops. These crops have a short period of resting. Then, When WSs are replaced or removed, labor and disposal costs are incurred. Even though WS are inexpensive, it is not appropriate to make them disposable in consideration of the SDGs.
Greenhouse cultivation has been increasing in Japan in recent years. Indoor cultivation (e.g., Venlo greenhouse), where crops can be grown all year round, needs to be operated for a long period. And the inside of the house is hot and humid and may be a harsher environment than outdoors. In other words, the inside of a greenhouse may need to be more durable than the outside.
In addition, the prices of the components of this WS are as follows. The microcomputer for communication (WioLTE) costs $ 110, the microcomputer for sensor measurement (Arduino Pro Mini) costs $ 120, the waterproof box costs $ 25, and the wind speed, wind direction, and rain gauge costs $ 120. When used in a place where power can be secured, the lithium polymer battery costs $ 25, and the power controller for the lithium polymer battery costs $ 20. When used in a place where power supply cannot be secured, the solar panel costs $ 100, the battery costs $ 120, the power controller costs $ 40, and the cable costs $ 30.
The price of these components is mostly the price of the WS itself. to make the price of the WS even cheaper, the microcomputer for communication (WioLTE) can be changed to a cheaper microcomputer and communication shield. However, there is no guarantee of stable communication.
If the WS is made cheaper, problems with sensor precision and waterproofing may occur. In addition, it may be difficult for the WS to operate stably and continuously for a long time with high precision. In many cases, it is difficult for the WS to operate continuously for even 7 months. Also, we cannot find a way to make the sensors and the box for WS cheaper.
From the above, price (low price) and continuous operation may not be proportional.
【Addition】Line 191-197
It is necessary to obtain stable weather information for a long time, more than one year. Crops grown outdoors include double-cropping and double-season crops with short fallow periods. Especially, indoor cultivation (e.g., Venlo greenhouse), where crops can be grown all year round, also needs to be operated for a long period. When WSs are replaced or removed, labor and disposal costs are incurred. Even though a WS is inexpensive, it is not appropriate to make them disposable in consideration of the SDGs.
- The problem of data gathering on opinions.
Lines 563-565. The authors mention "the opinions of agricultural workers and Japan Agricultural Cooperatives Zen-Noh Yamagata (JA) staff" to guide the development of User Interface (UI) for potential recipients of information from the KOSEN device. However, there is no explanation of how the opinions were gathered, including the instrument to gather the opinions, the validation of the opinions, and the synthesis of the opinions. It raises doubts whether those "opinions" are scientifically justified. The authors must add a specific subsection to explain the scientific process of gathering the "opinions," which includes the approach, instrument, validation, and synthesis/analysis process. This section could be added into the new "Methodology" section explaining the engineering design methodology.
【Reply】
As you mentioned, we describe the methodology of engineering design in the design of weather information provision. We explain in the following order: problem definition, opinion, instrument, validation, and synthesis/analysis.
【Addition】Line 465-521
3.4.2. Provision of weather information to farmers
・ Problem; In defining the problem in WS, we discussed what weather information farmers need, how to provide reliable information, how to provide information that is easy for farmers to understand, and how to prioritize the provision of weather information.
・ Opinions; To understand the current situation in the agricultural field and to obtain the various opinions of agricultural workers, we conducted a literature survey and a questionnaire survey of agricultural workers and JA staff.
The questionnaire items include crops grown by farmers, kinds of pull-type weather information, kinds of push-type weather information, frequently used places, and devices for information provision. We will summarize this information and discuss how to solve the problem in the following " Instrument ".
・Instrument; Based on the results of the above questionnaire survey and the knowledge obtained by the authors from the field observation, a webpage program for the weather information provision and an application for alert notification via LINE was developed.
Devices for providing information, kinds of pull-type information, and kinds of push-type information in weather information provision are summarized as follows.
The device for providing weather information to farmers is not a dedicated device, but a smartphone that is always carried by farmers. The application for providing weather information was then decided to be LINE, an application that is already installed on the smartphones of agricultural workers and is often used by them.
The items of pull-type information are temperature, humidity, solar radiation, wind speed, wind direction, and rainfall. However, for indoors (greenhouses), only temperature, humidity, and solar radiation are provided, since wind speed, wind direction, and rainfall are not necessary. Weather information for the past (one week) is also provided as needed. Push-type information includes alert notifications. If damage to crops occurs during high or low temperatures, an alert will be sent to the farmer's smartphone.
Based on the above, we have decided on the method of providing information. In the pull-type weather information provision, a webpage is created to provide weather information, and the most frequently used temperature and humidity are displayed at the top of the webpage. In the push-type weather information provision, alerts will be sent via LINE. We confirmed through the preliminary survey that it is possible to receive alerts via LINE. And then, to ensure that users are notified, a continuous alert notification is necessary.
・ Validation; We will verify whether the pull-type and push-type weather information could be reliably provided to farmers and JA staff. WSs were installed in the field and its weather information was provided to the farmers. Then, we conducted a questionnaire survey from the farmers.
Pull-type information provision and push-type information provision were validated through a questionnaire survey of agricultural workers and JA staff. The questionnaire items for the pull-type information provision include the readability of the webpage, the type of weather information, the size of the text, and the size of the graphs. The questionnaire items for push-type information provision were to determine whether the notifications were reliable, the number of notifications to keep sending until the user confirmed them, how to stop the alert notifications, and how to reset the alert notifications.
We summarized the results of a questionnaire survey on the provision of weather information. In the pull-type information provision, there were opinions that the text should be larger; the maximum and minimum temperatures of the previous day, today's minimum temperature, and the current temperature should be presented. In push-type information provision, there were opinions that notifications should be stopped after two confirmations of alert notifications, and that resetting alert notifications once a day would be sufficient.
・ Synthesis/Analysis; In synthesis and analysis process, the webpage and alert notifications were improved to reflect the questionnaire survey and the authors' knowledge. As a result, a good evaluation was obtained from farmers and JA staff.

Round 2
Reviewer 1 Report
The authors have addressed the previous concerns adequately.
Author Response
Reviewer 1
We sincerely appreciate the reviewers' comments.
Comments and Suggestions for Authors
English language and style are fine/minor spell check required
【Reply】
As you indicated, we have proofread the manuscript. Please check the manuscript.
Reviewer 2 Report
After a thorough check on the revised manuscript, I see that the authors have put effort into improving the quality of their manuscript. However, there are critical points the authors have not adequately addressed. The following points emerge as the basis of my recommendation.
Language concern.
- The language used in this manuscript seems unnatural, especially in the texts added as part of the revision process. The authors should consider having a moderate proofreading process over the added texts.
Engineering Design (lines 347-521).
- This research does not focus on developing an engineering design methodology. Therefore, the flow of engineering design taken by the authors (Problem Definition, Opinions, Instruments, Validation, and Synthesis/Analysis) should simply refer to an existing engineering design method (for example, VDI 2222/2221, Pahl & Beitz's, Dym & Little's, Cross', or any other established engineering design). The authors should clearly cite relevant references that support the design of the engineering process in this research. Otherwise, again, the engineering process of KOSEN Weather Station looks arbitrary and is not scientifically designed based on an established engineering design methodology.
- Since engineering design refers to the design of the engineering process (before the product development begins), the Methodology section should appear before the development process of KOSEN Weather Station. The authors should consider relocating it in between the "Related Work" section (currently Section 2) and the "KOSEN Weather Station and Provision of Weather Information to Farmers" section (currently Section 3).
- The engineering design should cover the entire engineering process in a linear flow. Therefore, the development of the technology (KOSEN Weather Station) and the target process (Provision of weather information to farmers) should be merged into one linear explanation. Looking at the current content of Sections 3.4.1 and 3.4.2, the content of Section 3.4.2 should be an integral part of Section 3.4.1. For each activity in Section 3.4.2, please merge it to its matched stage in section 3.4.1.
- Then, explanations of the engineering design process focus on the flow of the engineering process. In that sense, there should be no mention of primary information from the conduct of technology design (e.g., size of the box [lines 425-427], communication systems used [lines385-393], etc.). In short, the authors should separate the explanations on engineering design (Methodology) and the descriptions of detailed technical design (Design Process).
Design Process.
- All primary information (problem definition practices, the content and conduct of questionnaire surveys, detailed technical design, etc.,) should be moved and merged into the entire "Design Process" section. Consider merging lines 181-209, lines 377-434, lines 481-499, lines 504-518 into their matched stage in the entire Design Process (Sections "KOSEN Weather Station and Provision of Weather Information to Farmers", "Basic Experiments", "Field Experiments", or "Provision of Weather Information to Farmer").
- Again, regarding data gathering through "questionnaires" and "discussion", the authors should explain them in detail. Rather than sudden mentions of the results ("price should be less than USD 1,000"; "maintenance fee should less than USD 10", etc.), the authors should explain in detail the content of questionnaire/survey/discussion, the process, and how to synthesize the result of the survey/discussion. It will convince readers that every single process in this research is scientifically conducted.
I appreciate all efforts the authors have made to address the concerns of the reviewers. I would like to say a piece of good luck with the publication, and for the continuity of research on similar or other relevant topics.
Author Response
Reviewer 2
We sincerely appreciate the reviewers' comments.
Comments and Suggestions for Authors
Language concern.
- The language used in this manuscript seems unnatural, especially in the texts added as part of the revision process. The authors should consider having a moderate proofreading process over the added texts.
【Reply】
As you indicated, we have proofread the manuscript. Please check the manuscript.
Engineering Design (lines 347-521).
- This research does not focus on developing an engineering design methodology. Therefore, the flow of engineering design taken by the authors (Problem Definition, Opinions, Instruments, Validation, and Synthesis/Analysis) should simply refer to an existing engineering design method (for example, VDI 2222/2221, Pahl & Beitz's, Dym & Little's, Cross', or any other established engineering design). The authors should clearly cite relevant references that support the design of the engineering process in this research. Otherwise, again, the engineering process of KOSEN Weather Station looks arbitrary and is not scientifically designed based on an established engineering design methodology.
【Reply】
As you mentioned, we cited the following references in 2.2 Methodology.
[Ref 1] Pahl G, Beitz W, Feldhusen J, Grote K-H, Engineering Design - A Systematic Approach. Wallace K, Blessing L (Trans. and Eds.) 3rd ed. Springer, Berlin (2007).
[Ref 2] T. Tomiyama, P. Gul, Y. Jin, D. Lutters, C. Kind, F. Kimura, “Design Methodologies: Industrial and Educational Applications,” CIRP Ann. – Manuf. Technol., 58 (2009), pp.543-565. https://doi.org/10.1016/j.cirp.2009.09.003.
[Ref 3] Engineering Design Process,
https://www.teachengineering.org/populartopics/designprocess. [Accessed Feb. 25, 2022]
- Since engineering design refers to the design of the engineering process (before the product development begins), the Methodology section should appear before the development process of KOSEN Weather Station. The authors should consider relocating it in between the "Related Work" section (currently Section 2) and the "KOSEN Weather Station and Provision of Weather Information to Farmers" section (currently Section 3).
- The engineering design should cover the entire engineering process in a linear flow. Therefore, the development of the technology (KOSEN Weather Station) and the target process (Provision of weather information to farmers) should be merged into one linear explanation. Looking at the current content of Sections 3.4.1 and 3.4.2, the content of Section 3.4.2 should be an integral part of Section 3.4.1. For each activity in Section 3.4.2, please merge it to its matched stage in section 3.4.1.
【Reply】
As you mentioned, we have merged sections 3.4.1 and 3.4.2. We the moved 3.4 the methodology to the back of Related Studies.
【After correction】 2.2 Methodology
Please check the manuscript.
- Then, explanations of the engineering design process focus on the flow of the engineering process. In that sense, there should be no mention of primary information from the conduct of technology design (e.g., size of the box [lines 425-427], communication systems used [lines385-393], etc.). In short, the authors should separate the explanations on engineering design (Methodology) and the descriptions of detailed technical design (Design Process).
【Reply】
As you suggested, we have revised the explanations on engineering design (Methodology) and the descriptions of detailed technical design (Design Process) separately.
【After correction】 3.2 KOSEN weather station Line 380
The WS size should be less than 200 mm wide, less than 300 mm high, and less than 150 mm deep because the WS size should be sufficiently small for farmers to install easily.
【After correction】 3.2 KOSEN weather station Line 364
Consideration of connection methods between the microcomputer and the communication device is necessary. In general, the most common method is connection to the micro-computer using a USB connection or a shield. Many microcomputers also have a built-in communication function. It is necessary to consider compatibility between communication devices and the microcomputer, and corrosion in high-temperature and humid environments. Microcomputers with built-in communication functions are more expensive, but they can operate stably and continuously. They are useful for a long time, even in hot and humid environments. We discussed communication systems. The most used communication systems are Cellular networks and Low Power Wide Area (LPWA) networks. Cellular networks (4G and 3G) have a wide communication range and a low initial cost. However, Communication devices using cellular networks consume much power and have running costs. LPWAs include ZigBee, LoRa, Sigfox, and so on. Communication de-vices using LPWA have low power consumption and low running cost. However, LPWA might cause communication instability because of crop growth. In addition, the initial cost is highly attributable to the need for a repeater nearby to transmit weather information to the server.
Design Process.
- All primary information (problem definition practices, the content and conduct of questionnaire surveys, detailed technical design, etc.,) should be moved and merged into the entire "Design Process" section. Consider merging lines 181-209, lines 377-434, lines 481-499, lines 504-518 into their matched stage in the entire Design Process (Sections "KOSEN Weather Station and Provision of Weather Information to Farmers", "Basic Experiments", "Field Experiments", or "Provision of Weather Information to Farmer").
【Reply】
We have integrated your suggestions into their matched stage in the entire Design Process.
Please check the manuscript.
【After correction】 2.1 Related work Line 179
Based on the explanation presented above, this paper presents simultaneous discussion of the WS development, server construction, and weather information provision. To develop a WS that is useful by small-scale and medium-scale farmers, we consider the sensor, microcontroller for sensor measurement, communication function, circuit board design, power supply, power consumption, WS size, and price. To store the weather information uploaded from the WS and to provide it to farmers thereafter, we must also build a server. This system has been developed according to an engineering design for which the details are described in the next section.
【After correction】 3.2 KOSEN weather station Line 330
As described in this paper, the types of weather information to be provided to farmers were ascertained through a questionnaire survey of farmers and JA staff; six types of weather information were determined.
【After correction】 3.2 KOSEN weather station Line 352
The circuit board design is described. The circuit diagram and circuit board for the sensor unit are portrayed in Fig. 4. A durable circuit board that is useful for long periods of time in hot and humid indoor and outdoor environments must be designed. Printed circuit boards (PCBs) and printed circuit board assemblies (PCBAs) are recommended for stable measurement over a long period of time. Actually, PCBs are inexpensive, but they require soldering, and require countermeasures against corrosion, etc. Although PCBAs are more expensive, they require no soldering and are more durable. Eventually, PCBs were chosen because of their price, particularly considering that farmers operating on a small scale or medium scale use them. Furthermore, the wiring was designed to be sufficiently thick to withstand overcurrent, etc.
【After correction】 3.2 KOSEN weather station Line 382
Final, the power supply is described. To use the WS for a long time even outdoors, low power consumption is preferred. In general, the microcomputer power consumption is highest during communication. Supplying sufficient power is necessary, even when using cellular networks (4G and 3G). One must also consider where power can be secured and where it is unavailable. In places where power is available, we can use a USB adapter. By contrast, in places where power is not available, solar panels and batteries must be used considering the power consumption of a WS. Because Japan has a rainy season, it is necessary to secure a power source that is useful during this period. The amount of power generated by the solar panel and the battery capacity must also be considered. Furthermore, this WS can switch easily between the two power sources. In places where the power supply cannot be secured, the WS uses a power controller [44] for solar panels and automobile batteries.
【After correction】 3.3 Cloud server Line 438
The device for providing weather information to farmers is not a dedicated device, but is rather a smartphone that is always carried by farmers. Dedicated devices can provide good information, but they are expensive and take time to develop. The application for providing weather information was then decided to be LINE, an application that is already installed on the smartphones of agricultural workers and is often used by them. Pull-type and push-type modes are available for information provision. The pull-type information provision is a method that allows farmers to access a web page and browse weather information when users want to check it. The pull-type information items are temperature, humidity, solar radiation, wind speed, wind direction, and rainfall. However, for indoors (greenhouses), only temperature, humidity, and solar radiation are provided because wind speed, wind direction, and rainfall are not necessary. Weather information for the past (one week) is also provided as needed. Push-type information includes alert notifications. The push-type information provision is a method that presents information from a server, irrespective of the agricultural worker’s intention. For example, an alert notification is sent irrespective of the user's intention when the weather conditions can cause damage to the crop.
【After correction】 3.3 Cloud server Line 453
Using the cloud server entails a running cost. It costs approximately $10 or less on average per WS.
【After correction】 5.2 Experiment results Line 607
The authors conducted a questionnaire survey of farmers and JA staff members regarding sensor precision, WS size, and ease of installation. As a result, they gave good evaluations.
【After correction】 5.3 Discussion Line 622
The solar panel output used for this study is 100 W. The automobile battery capacity is 100 Ah. Furthermore, in places where the power supply can be secured, the WS uses the power controller [48] for a LiPo battery to supply the power stably.
- Again, regarding data gathering through "questionnaires" and "discussion", the authors should explain them in detail. Rather than sudden mentions of the results ("price should be less than USD 1,000"; "maintenance fee should less than USD 10", etc.), the authors should explain in detail the content of questionnaire/survey/discussion, the process, and how to synthesize the result of the survey/discussion. It will convince readers that every single process in this research is scientifically conducted.
【Reply】
As you pointed out, we also explained in detail the content of the survey and discussion, the process, and how to summarize the results.
【After correction】 3.2 KOSEN weather station Line 394
A questionnaire survey was administered among workers on small-scale and medium-scale farms and JA staff on WS prices. It elicited various opinions from agricultural workers. However, after discussing commercialization of this system, we came to the conclusion that the initial cost of this WS is expected to be less than $1,000 and that the running cost is expected to be less than $10. The KOSEN-WS for outdoor installation, which re-quires a solar panel and battery, costs approximately $ 950, and the WS for outdoor installation, which uses a commercial power source, costs approximately $ 550 because it does not require a solar panel and battery. WS for indoor installation, which uses a commercial power source, costs approximately 480 $ because it does not require sensors for rainfall, wind speed, and wind direction.
【After correction】 6. Provision of Weather Information to Farmer Line 714
The running cost of the cloud servers came to about $10 per WS each month. We received good feedback from farmers and JA staff that it was an appropriate fee.